# TBC1D9 regulates TBK1 activation through Ca$^{2+}$ signaling in selective autophagy

Takashi Nozawa [1], Shunsuke Sano[1], Atsuko Minowa-Nozawa[1], Hirotaka Toh[1], Shintaro Nakajima [2,3], Kazunori Murase[1], Chihiro Aikawa[1] & Ichiro Nakagawa [1]*

Invading microbial pathogens can be eliminated selectively by xenophagy. Ubiquitin-mediated autophagy receptors are phosphorylated by TANK-binding kinase 1 (TBK1) and recruited to ubiquitinated bacteria to facilitate autophagosome formation during xenophagy, but the molecular mechanism underlying TBK1 activation in response to microbial infection is not clear. Here, we show that bacterial infection increases Ca$^{2+}$ levels to activate TBK1 for xenophagy via the Ca$^{2+}$-binding protein TBC1 domain family member 9 (TBC1D9). Mechanistically, the ubiquitin-binding region (UBR) and Ca$^{2+}$-binding motif of TBC1D9 mediate its binding with ubiquitin-positive bacteria, and TBC1D9 knockout suppresses TBK1 activation and subsequent recruitment of the ULK1 complex. Treatment with a Ca$^{2+}$ chelator impairs TBC1D9–ubiquitin interactions and TBK1 activation during xenophagy. TBC1D9 is also recruited to damaged mitochondria through its UBR and Ca$^{2+}$-binding motif, and is required for TBK1 activation during mitophagy. These results indicate that TBC1D9 controls TBK1 activation during xenophagy and mitophagy through Ca$^{2+}$-dependent ubiquitin-recognition.

[1] Department of Microbiology, Graduate School of Medicine, Kyoto University, Yoshida-Konoe-cho, Sakyo-ku, Kyoto 606-8501, Japan. [2] Department of Life Science Dentistry, The Nippon Dental University, Tokyo 102-8159, Japan. [3] Department of Developmental and Regenerative Dentistry, School of Life Dentistry at Tokyo, The Nippon Dental University, Tokyo 102-8159, Japan. *email: nakagawa.ichiro.7w@kyoto-u.ac.jp

Macroautophagy (i.e. autophagy) is an intracellular degradation pathway that involves delivery of cytoplasmic materials, such as cytosolic components, organelles, and invading microbes, to lysosomes and thereby has important functions in cellular homeostasis, stress response, and nutrient recycling[1,2]. Autophagy is initiated by the formation of an isolation membrane, which elongates to engulf target materials and becomes a double-membrane vacuole called an autophagosome. Autophagosomes can either nonselectively degrade cytoplasmic components during nutrient starvation or selectively sequester unwanted materials, including microbial pathogens or damaged organelles, using autophagic receptors that tag autophagy targets and bind to microtubule-associated protein 1 light chain 3 (LC3) in autophagosomes[3–6].

Autophagy receptors are divided into two groups depending on the targeting mechanism: ubiquitin-binding receptors and organelle-resident receptors[7]. Ubiquitin-binding receptors, such as p62/sequestosome 1, optineurin (OPTN), and nuclear dot protein 52 (NDP52), are involved in selective autophagy against microbes (xenophagy), depolarized mitochondria (Parkin-mediated mitophagy), and protein aggregates (aggrephagy)[3,4,8]. We previously reported that the organelle-resident RAB35 GTPase promotes NDP52 recruitment to ubiquitin-tagged targets[9], and recent studies show that TANK-binding kinase 1 (TBK1) regulates ubiquitin-binding autophagy receptors associated with several types of selective autophagy[10,11]. TBK1 can phosphorylate p62, OPTN, and NDP52 to promote selective autophagy by facilitating their interaction with LC3, ubiquitin, and RAB35, respectively[9,11,12]. OPTN and NDP52, in turn, facilitate TBK1 activation during xenophagy and mitophagy[9,13]. NDP52 is also involved in the targeting and activation of the ULK1 complex during selective autophagy[14,15].

Tre-2/Bub2/Cdc16 (TBC)/RabGTPase-activating proteins (GAPs) are important regulators of intracellular membrane trafficking, including autophagy, and several TBC/RabGAPs, including TBC1 domain family member (TBC1D)5, TBC1D10A, TBC1D14, and TBC1D25, regulate starvation-induced autophagy[9,16–18], whereas TBC1D15 and TBC1D17 are required for mitophagy[19], and TBC1D10A is involved in xenophagy and mitophagy[9]. However, during these processes, the biological functions of TBC/RabGAPs except for RabGTPase-activation have not been elucidated.

TBK1 is an important regulator of innate immune responses. Upon infection by DNA viruses or bacteria, microbial DNA delivered into the cytosol stimulates the production of type I interferon (IFN) through stimulator of IFN gene (STING), which triggers phosphorylation of IFN-regulatory factor 3 (IRF3) by TBK1 to promote nuclear translocation and transcriptional activation of genes, including IFNβ[20–22]. Additionally, STING-mediated TBK1 activation is required for autophagy in response to *Mycobacterium tuberculosis* DNA[23], indicating that a DNA-sensing pathway could prime xenophagy. On the other hand, other types of selective autophagy, including mitophagy and lysophagy, also involve TBK1; however, the molecular mechanism underlying TBK1 activation in response to microbial infection or organelle damage remains to be established[11,13,14,24].

In this study, we confirm the involvement of a DNA-sensing pathway in TBK1 activation using *STING*-knockout (KO) cells during infection of Group A *Streptococcus* (GAS), a major bacterial pathogen and target of xenophagy, and show that a STING-mediated pathway is not involved in TBK1 activation during GAS infection. We also perform overexpression screening of RabGAPs involved in TBK1 activation, and identify TBC1D9 as a regulator of TBK1-mediated autophagy. We show that cytosolic $Ca^{2+}$ signaling is required for TBK1 activation during xenophagy and mitophagy and this process is regulated by $Ca^{2+}$-binding TBC1D9,

highlighting TBC/RabGAP-mediated regulation of TBK1 activation in selective autophagy.

## Results

**TBC1D9 is involved in TBK1 phosphorylation**. We previously reported that GAS internalized via endocytosis enters the cytosol by secreting streptolysin O (SLO), a pore-forming toxin, and autophagosome formation in response to cytosolic GAS is induced through an SLO-dependent mechanism[25]. To investigate whether TBK1 activation is also triggered by SLO, we infected cells with GAS wild-type (WT) and isogenic SLO mutants (Δ*slo*) and examined TBK1 (S172) phosphorylation. GAS WT infection resulted in rapid TBK1 activation according to increased phosphorylation at TBK1 (S172), whereas TBK1 was not activated during Δ*slo* mutant infection (Supplementary Fig. 1a), demonstrating that TBK1 activation is induced in response to GAS invasion into the cytosol and/or endosomal membrane damage by SLO.

A previous study suggests that the intracellular DNA sensor cyclic GMP–AMP synthase and STING lead to TBK1 activation via phosphorylation at S172 in response to viral or bacterial infection[26]. This DNA-sensing pathway is critical for IFNβ production and autophagy against invading *M. tuberculosis*[23]. Here, we generated *STING*-KO HeLa cells and infected them with GAS and the DNA virus herpes simplex virus-II (HSV-II). TBK1 activation in response to HSV-II infection was completely suppressed in *STING*-KO cells, whereas TBK1 activation during GAS infection was unaffected by *STING*-KO (Supplementary Fig. 1b). Additionally, TBK1 activation in the DNA-sensing pathway requires an intact Golgi apparatus, and treatment with brefeldin A (BFA), which blocks endoplasmic reticulum (ER)-to-Golgi traffic[27], abolishes STING-dependent phosphorylation of TBK1 (refs. [28,29]). Our results showed that TBK1 activation during GAS infection was not inhibited by BFA treatment (Supplementary Fig. 1c), suggesting TBK1 activation during GAS infection does not require an intact Golgi apparatus. Moreover, phosphorylation of the TBK1 substrate IRF3 is induced upon HSV-II infection; however, we did not observe this during GAS infection (Supplementary Fig. 1d). These results suggested that the TBK1-activation pathway during GAS infection differs from the STING/TBK1-mediated DNA-sensing pathway.

We previously identified TBC1D10A and its substrate RAB35 as regulators of autophagy, with RAB35 involved in TBK1 activation possibly through NDP52 recruitment[9]. We hypothesized that other TBC/RabGAPs might be involved in TBK1 activation in xenophagy. To test this hypothesis, we performed comprehensive screening for TBC/RabGAPs that modulate TBK1 activation during GAS infection. We engineered HeLa cells overexpressing emerald green fluorescent protein (EmGFP)-tagged TBC/RabGAPs and after GAS infection, we quantified the ratio of phosphorylated TBK1 (p-TBK1) to total TBK1. Of the 26 TBC/RabGAPs tested, overexpression of TBC1D10A, TBC1D2, TBC1D8, and TBC1D18 suppressed TBK1 activation, whereas overexpression of TBC1D9 and TBC1D24 significantly promoted TBK1 activation (Supplementary Fig. 2a, b). Given that TBC1D9 colocalized with the autophagosome during GAS infection[9], we focused on the roles of TBC1D9 in TBK1-mediated xenophagy.

TBC1D9 is expressed in various tissues; however, its function remains unknown[30]. To examine TBC1D9 involvement in TBK1 activation, we constructed *TBC1D9*-KO HeLa cells by CRISPR/Cas9 genome editing (Supplementary Fig. 2c). We found that GAS-infection-induced TBK1 phosphorylation increased over time in WT HeLa cells, whereas this increase was significantly diminished in *TBC1D9*-KO cells (Fig. 1a, b). Visualization of

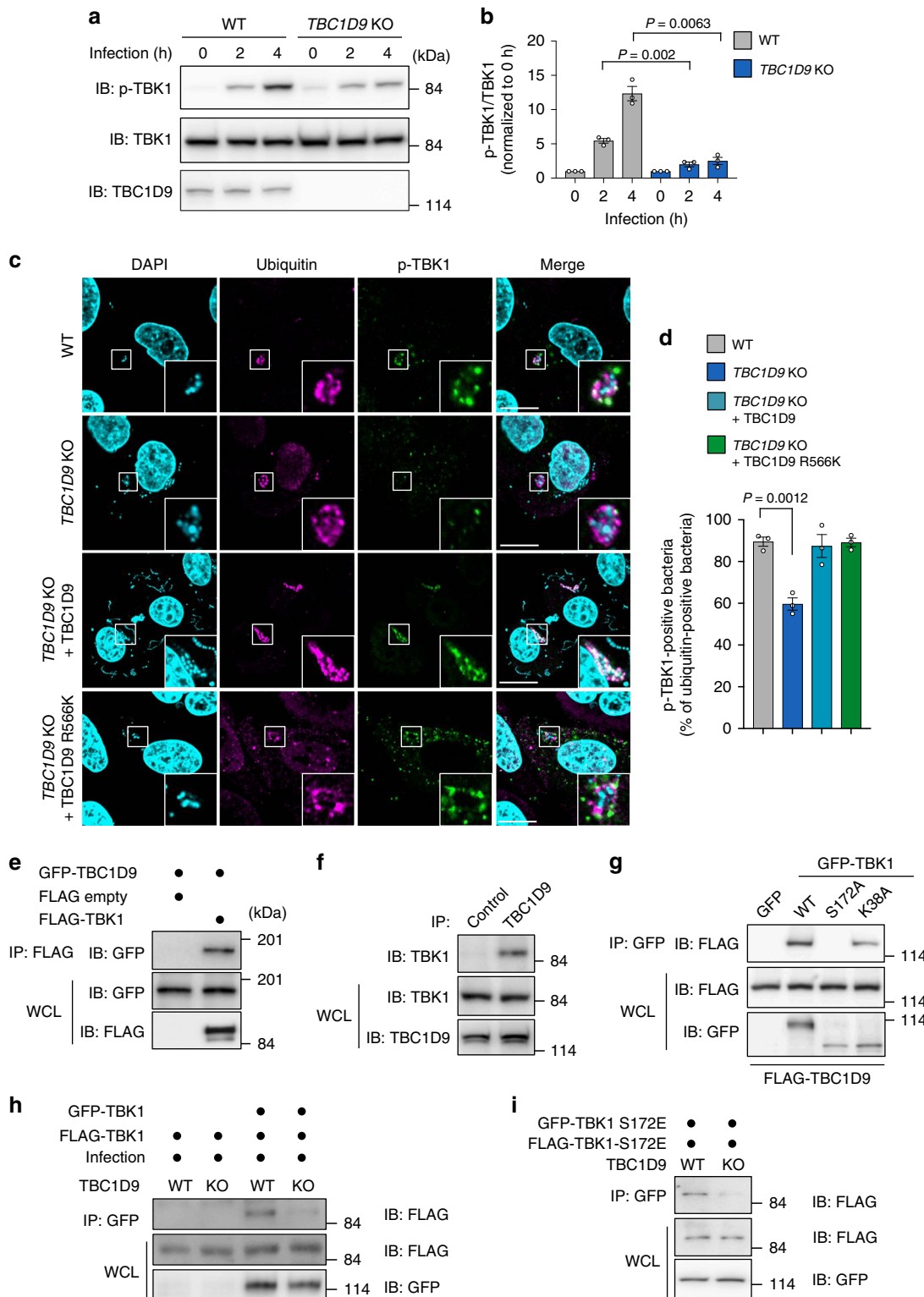

**Fig. 1 TBC1D9 is involved in TBK1 activation independent of GAP activity. a**, **b** Western blots of WT or *TBC1D9*-KO cells during GAS infection. **a** HeLa WT or *TBC1D9*-KO cells were infected with GAS for the indicated times and analyzed by western blot. **b** Quantification of the p-TBK1/TBK1 ratio. **c**, **d** HeLa WT, *TBC1D9*-KO, *TBC1D9*-overexpressing *TBC1D9*-KO, and *TBC1D9 R566K*-overexpressing *TBC1D9*-KO cells were infected with GAS for 4 h, fixed, and immunostained for ubiquitin (magenta) and p-TBK1 (green). Cellular and bacterial DNA were stained with DAPI (cyan). **c** Representative confocal images and **d** quantification of p-TBK1-positive ubiquitin-coated GAS. **e** Coimmunoprecipitation of GFP-TBC1D9 and FLAG-TBK1. **f** Coimmunoprecipitation of endogenous TBC1D9 and TBK1. **g** Coimmunoprecipitation of GFP-TBK1 WT, S172A, K38A, and FLAG-TBC1D9. **h** Coimmunoprecipitation of GFP-TBK1 and FLAG-TBK1 in GAS-infected HeLa WT or *TBC1D9*-KO cells. **i** Coimmunoprecipitation of GFP-TBK1-S172E and FLAG-TBK1-S172E in HeLa WT or *TBC1D9*-KO cells. Scale bars, 10 μm. Data in (**b**) and (**d**) (*n* > 200 cells per condition) represent the mean ± SEM of three independent experiments. *P* values calculated by two-tailed Student's *t* test.

phosphorylated-TBK1 (p-TBK1) by confocal microscopy in WT cells revealed that 90% of ubiquitin-positive GAS was coated with p-TBK1, whereas p-TBK1-positive GAS was still observed but this percentage dropped to 54% in *TBC1D9*-KO cells (Fig. 1c, d). Moreover, expression of FLAG-TBC1D9 increased p-TBK1-positive GAS in *TBC1D9*-KO cells (Fig. 1c, d), suggesting that TBC1D9 is involved in TBK1 phosphorylation and targeting of invading GAS. To test if the GAP activity of TBC1D10A is necessary to activate TBK1, we constructed TBC1D9 mutants, in which a conserved arginine required for GAP activity had been replaced with lysine (R566K). TBC1D9 R566K fully rescued the recruitment of p-TBK1 to invading GAS in *TBC1D9*-KO cells (Fig. 1c, d), indicating that GAP activity is not required for TBK1 activation during GAS infection.

NDP52 and OPTN interact with TBK1 and are involved in TBK1 activation during mitophagy and xenophagy[13,31,32]. Immunoprecipitation assays revealed that both transiently expressed and endogenous TBC1D9 interaction with TBK1 (Fig. 1e, f). Additionally, we found that TBC1D9 interacted with a kinase dead mutant (TBK1 K38A), but did not interact with a nonphosphorylated mutant (TBK1 S172A) (Fig. 1g), suggesting that TBC1D9 specifically binds to p-TBK1.

We then investigated how TBC1D9 promotes TBK1 activation. Because TBK1 activation requires TBK1 oligomerization in order to allow trans-autophosphorylation, we examined whether TBK1 self-association involves TBC1D9. Immunoprecipitation assays showed that FLAG-TBK1 precipitated with GFP-TBK1 in WT cells but not in *TBC1D9*-KO cells (Fig. 1h). Self-association of the phospho-mimic TBK1 was also attenuated in *TBC1D9*-KO cells (Fig. 1i). Collectively, these results suggested that TBC1D9 facilitates TBK1 self-association to promote TBK1 phosphorylation.

**TBC1D9 is required for TBK1-mediated xenophagy**. To identify the role of TBC1D9 in the process of TBK1 activation during GAS infection, we examined the recruitment of known autophagy regulators against GAS in *TBC1D9* KO cells. We found that recruitment of RAB35 (ref.[9]), ubiquitin, galectin-3 (ref.[33]), and nucleotide-binding oligomerization domain-containing protein 2 (NOD2)[34,35] were unaffected by *TBC1D9* KO, whereas that of NDP52, p62, and LC3 was significantly reduced (Fig. 2a, b), suggesting that TBC1D9 is involved in autophagosome formation. To confirm whether TBC1D9 is involved in autophagosome formation, we examined the conversion of LC3-I to LC3-II during infection. Although LC3-II was increased in response to starvation in *TBC1D9*-KO cells, it was not changed during GAS infection in *TBC1D9*-KO cells (Fig. 2c, d), suggesting that TBC1D9 is required for autophagosome formation against GAS infection. These results suggest the involvement of TBC1D9 in TBK1 activation, followed by autophagy adaptor recruitment and autophagosome formation during GAS infection.

Recent advances have revealed that TBK1 and NDP52 recruit the ULK1 complex to cytosolic bacteria to initiate xenophagy[15,36]. To examine if TBC1D9 is also required for the recruitment of ULK1 to the invading GAS, we observed the ULK1 localization during infection. We found that mClover-ULK1 surrounded ubiquitin-positive GAS in WT cells, whereas this localization was significantly decreased in *TBC1D9*-KO cells (Fig. 2e, f). Intracellular survival of GAS at 6 h after infection was increased in ULK1-depleted cells (Supplementary Fig. 3a, b), indicating that ULK1 is required to eliminate the intracellular GAS. We also found that other ULK1 complex components (FIP200, ATG13, and ULK2) were similarly needed to degrade intracellular GAS (Supplementary Fig. 3a, b). Since bacterial survival was significantly increased by TBC1D9 knockout (Fig. 2g), it was

concluded that TBC1D9 is functionally relevant in xenophagy and its effects are exerted by promoting TBK1 activation and ULK1 complex recruitment.

**TBC1D9 is recruited to GAS via Ca$^{2+}$ and ubiquitin binding**. To understand the regulatory mechanism associated with TBC1D9 in xenophagy, we detected TBC1D9 subcellular localization during GAS infection. In response to GAS infection, EmGFP-TBC1D9 accumulated around intracellular GAS and colocalized with mCherry-LC3 (Fig. 3a). Moreover, TBC1D9 recruitment increased over time after infection, with the kinetics revealing that TBC1D9 localization to GAS was similar to that of ubiquitin and LC3 (Fig. 3b). To determine whether TBC1D9 targets cytosolic GAS, we detected the localization of endogenous TBC1D9 during GAS WT or Δ*slo* infection. As shown in Fig. 3c, 22.7% of WT GAS-infected cells showed endogenous TBC1D9-positive bacteria, which were rarely observed following Δ*slo* infection (Fig. 3c). Furthermore, we found that TBC1D9 was recruited to GAS, even in *autophagy related 5* (*ATG5*)-KO cells (Fig. 3d), suggesting TBC1D9 translocation proximal to invading bacteria prior to autophagosome formation.

Human TBC1D9 contains two GRAM domains (aa 146–213, and aa 293–361), a TBC domain (aa 515–702), and an EF-hand motif (aa 886–921) (Fig. 3e). Therefore, to identify the TBC1D9 region required for its recruitment to invading bacteria, we examined GAS-mediated recruitment of TBC1D9 deletion mutants. GFP-TBC1D9 Δ2–722 (lacking two GRAM domains and the TBC domain) was effectively recruited to GAS and colocalized with autophagosomes (Fig. 3f); however, mutants lacking the EF-hand motif (Δ723–1266 and ΔEF) displayed impaired recruitment. Additionally, despite containing the EF-hand motif, EmGFP-TBC1D9 Δ926–1266 did not localize to GAS. These results suggested that TBC1D9 is recruited to invading GAS through both the EF-hand motif and the region from aa 926 to aa 1100, and GAP activity is not required for the recruitment of TBC1D9.

Upon noting that TBC1D9 was only recruited to ubiquitin-positive GAS, we hypothesized that the region from aa 926 to aa 1100 binds to ubiquitin, which promotes recognition of target bacteria. The results of a glutathione *S*-transferase (GST) pulldown assay showed that TBC1D9 interacted with ubiquitin but not with ubiquitin-like proteins in vitro (Fig. 3g). We subsequently validated TBC1D9 interaction with polyubiquitin in cells using a proximity ligation assay (PLA), which revealed a TBC1D9–ubiquitin association in WT HeLa cells but not in *TBC1D9*-KO cells (Fig. 3h, i). Moreover, the PLA signals were significantly increased during GAS infection according to an SLO-dependent mechanism, with dense TBC1D9 accumulation proximal to invading WT GAS (Fig. 3h). These observations suggested that TBC1D9–ubiquitin interactions were enhanced in response to GAS invasion into cytosol via SLO and mainly occurred in the vicinity of cytosolic GAS.

Lys63-linked ubiquitination is reportedly important for xenophagy[37]. In GAS-infected cells, we observed accumulation of Lys48-linked ubiquitin chains around intracellular GAS as well as Lys63-linked ubiquitination (Supplementary Fig. 4). We then performed pulldown assays targeting the TBC1D9 aa 926 to aa 1100 region and Lys48- or Lys63-linked polyubiquitin. As shown in Fig. 3j, Lys63-linked polyubiquitin preferentially precipitated with this TBC1D9 variant, suggesting that the region aa 926 to aa 1100 interacted with the Lys63-linked ubiquitin chain. To confirm this, we detected interactions between EmGFP-TBC1D9 constructs and Lys63-linked polyubiquitin during GAS infection by PLA assay. As expected, deletion of aa 925 to aa 1100 significantly decreased PLA signals (Fig. 3k, l), indicating

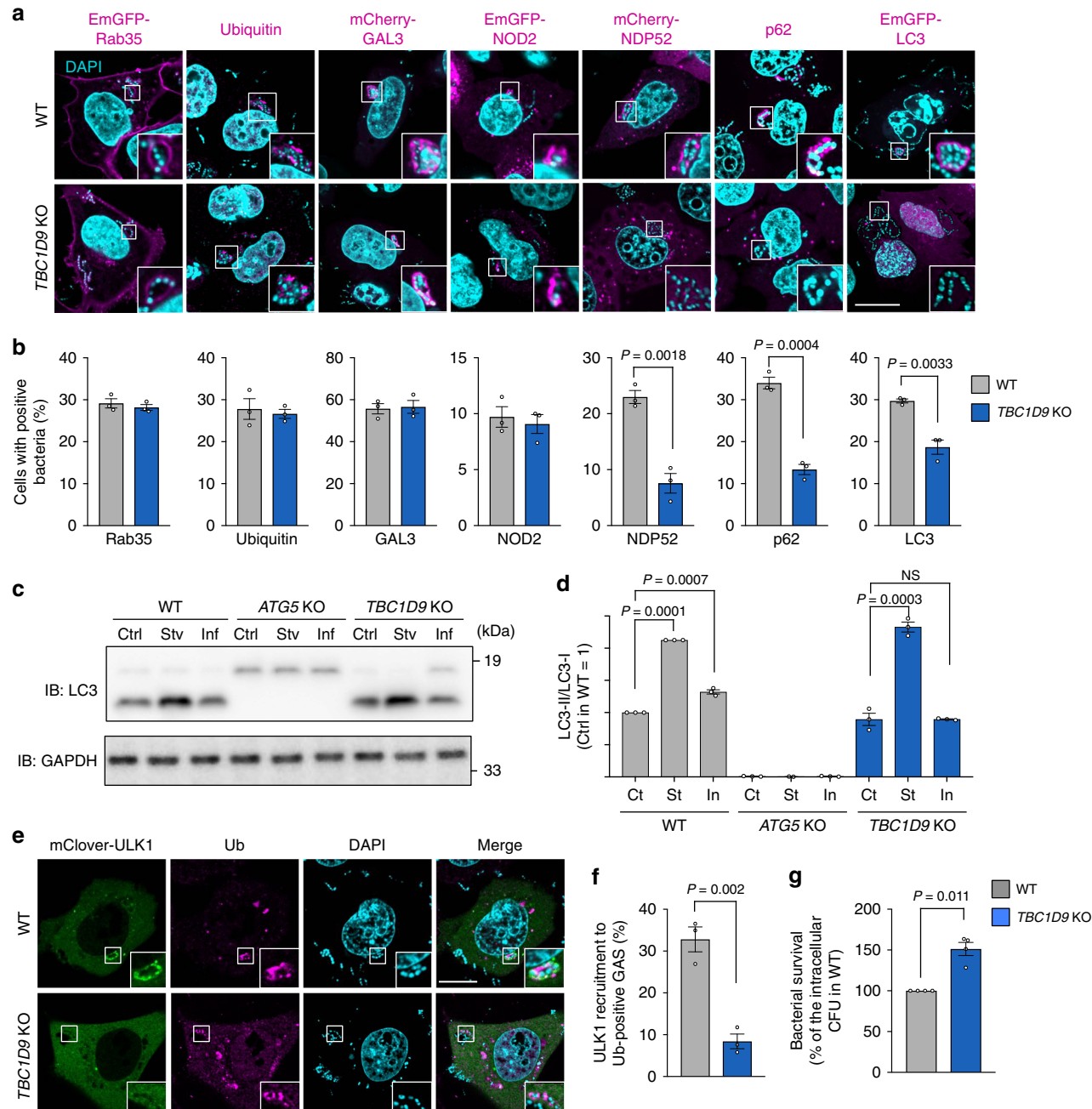

**Fig. 2 TBC1D9 is involved in xenophagy during GAS infection. a, b** Recruitment of the indicated proteins to invading GAS. **a** Representative confocal images and **b** quantification of GAS-positive cells. **c, d** Conversion of LC3-I to LC3-II. **c** Representative western blot of LC3 during starvation and GAS infection and **d** quantification of LC3-II/LC3-I ratio. **e, f** Localization of ULK1 in GAS-infected cells. **e** Representative confocal images of ULK1 recruitment to ubiquitin-coated GAS and **f** quantification of ULK1-positive GAS. **g** Intracellular bacterial CFU at 6 h p.i in WT and *TBC1D9*-KO HeLa cells. Scale bars, 10 μm. Data in (**b**) (*n* > 200 cells per condition), (**d, f**) (*n* > 50 bacteria per condition), and (**g**) represent the mean ± SEM of three independent experiments. *P* values calculated by two-tailed Student's *t* test.

that this region is required for the interaction with ubiquitin. These findings suggested that TBC1D9 was recruited to invading GAS through its ubiquitin-binding region (UBR; aa 926–1100) and the EF-hand motif, which promote binding to a Lys63-linked ubiquitin-chain.

**Ca$^{2+}$ mobilization is central to TBC1D9-mediated TBK1 activation.** Given that TBC1D9 is recruited to invading GAS via the EF-hand motif, which is a Ca$^{2+}$-binding motif, and

because cytosolic Ca$^{2+}$ concentrations are low under normal physiological conditions, we investigated elevations in Ca$^{2+}$ levels during GAS infection. We expressed G-CaMP, a genetically encoded Ca$^{2+}$ indicator, in HeLa cells and subsequently infected them with GAS or treated them with the calcium ionophore A23187. We observed that the average intensity of G-CaMP increased over time during GAS infection in an SLO-dependent manner, with the average G-CaMP intensity at 4 h comparable with that in A23187-treated cells (Supplementary Fig. 5a, b) and indicating GAS internalization into the cytosol and/or endosomal

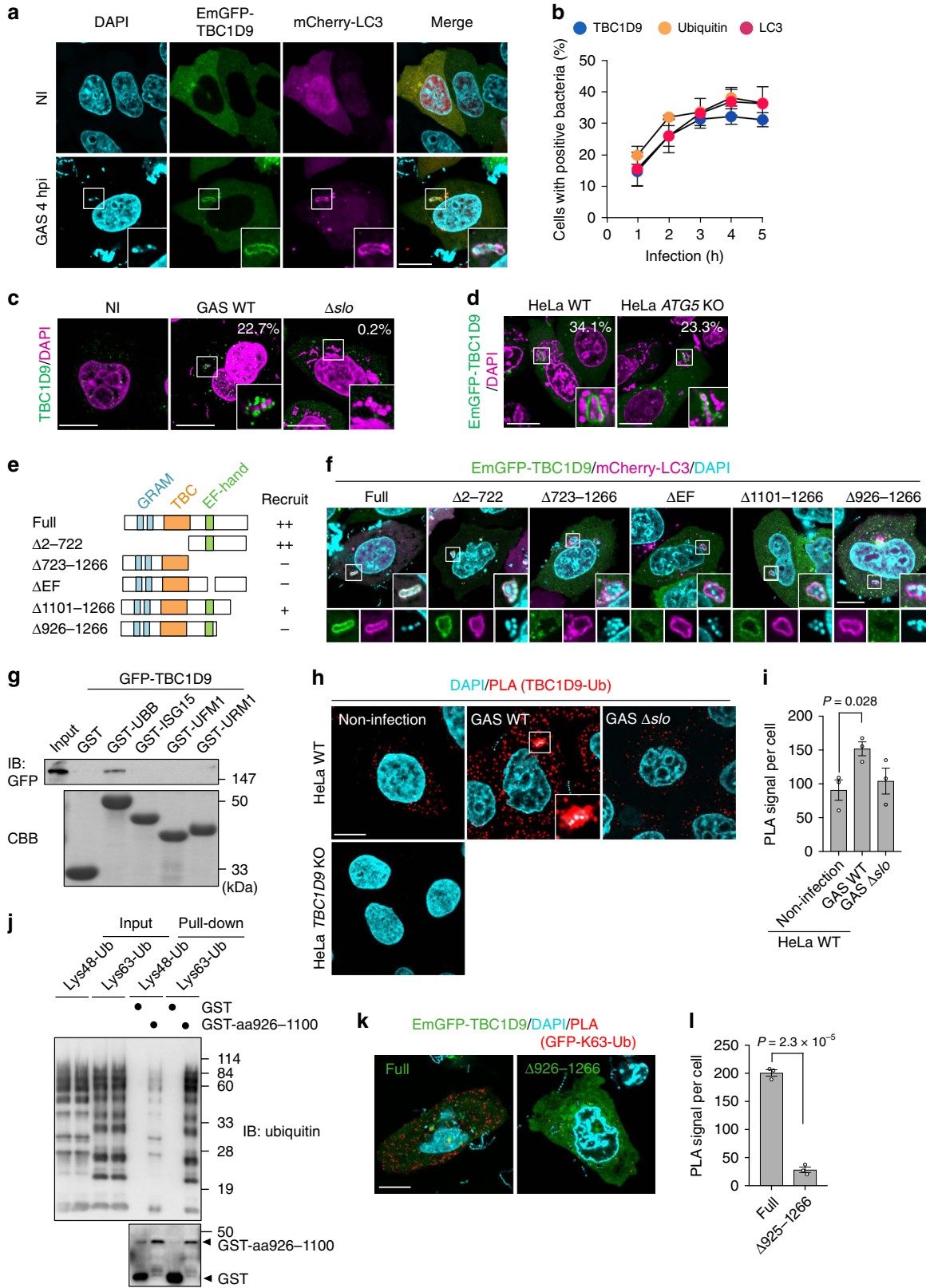

membrane damage via SLO-induced cytosolic $Ca^{2+}$ mobilization. We validated $Ca^{2+}$ elevation using the ratiometric $Ca^{2+}$ indicator, Fura 2-AM, in order to rule out changes in cell volume or other artifacts occurring upon GAS infection. Fura 2-AM ratio kinetics revealed that cytosolic $Ca^{2+}$ increased from 2 h after infection in an SLO-dependent mechanism (Fig. 4a).

We then examined the effects of the intracellular $Ca^{2+}$ chelator BAPTA-AM on TBC1D9 and LC3 recruitment to GAS, finding a substantial reduction in this activity following BAPTA-AM treatment (Fig. 4b–d). These results suggested that elevated $Ca^{2+}$ levels during GAS infection induced TBC1D9 and LC3 recruitment. We then examined ubiquitin, p62, and RAB35

**Fig. 3 TBC1D9 recruitment to invading GAS requires its Ca$^{2+}$-binding site and UBR. a, b** TBC1D9 recruitment to GAS. HeLa cells expressing GFP-TBC1D9 and mCherry-LC3 were infected with GAS, fixed, and stained with DAPI. **a** Representative confocal images of GFP-TBC1D9 localization in uninfected or GAS-infected cells and **b** the time course of TBC1D9, ubiquitin, and LC3 recruitment during GAS infection. **c** HeLa cells were infected with GAS WT or the Δ*slo* mutant for 4 h, fixed, and immunostained for TBC1D9. The percentage of TBC1D9-positive GAS-infected cells is shown. **d** WT or *ATG5*-KO cells expressing GFP-TBC1D9 were infected with for 4 h, and the percentage of TBC1D9-positive GAS-infected cells is shown. **e** Domain organization and deletion mutants of TBC1D9. **f** Localization of GFP-TBC1D9 deletion mutants during GAS infection. **g** Beads coated with GST, GST-ubiquitin, or ubiquitin-like proteins were incubated with lysate from cells expressing GFP-TBC1D9, followed by immunoblot with an antibody against GFP. **h**, **i** HeLa cells infected with GAS WT or the Δ*slo* mutant were stained with TBC1D9 and primary antibodies against Lys63-specific ubiquitin to assess TBC1D9-ubiquitin binding by Duolink PLA. **h** Representative confocal micrographs and **i** quantification of the PLA signal per cell. Dots (red) indicate TBC1D9-ubiquitin complexes. Scale bars, 10 μm. **j** GST pulldown assay with indicated TBC1D9 construct and GST as a control. GST-fusion proteins were incubated with purchased diubiquitins linked by Lys48 or Lys63 (Boston Biochem) and analyzed by immunoblotting with antiubiquitin antibody. **k**, **l** HeLa cells expressing EmGFP-TBC1D9 constructs were infected with GAS, fixed, and stained with primary antibodies against anti-GFP and anti-Lys63-specific ubiquitin to assess TBC1D9-ubiquitin binding by Duolink PLA. **k** Representative confocal micrographs and **l** quantification of the PLA signal per cell. Data in (**b**) (*n* > 200 cells per condition), (**c**) (*n* > 200 cells), (**d**) (*n* > 200 cells), (**i**) (*n* > 20 cells per condition), and (**l**) (*n* > 20 cells per condition) represent the mean ± SEM of three independent experiments. *P* values calculated by two-tailed Student's *t* test.

localization to GAS in BAPTA-AM-treated cells. We found that BAPTA-AM treatment did not affect ubiquitin accumulation and RAB35 recruitment to intracellular GAS but significantly reduced p62 recruitment to ubiquitin-positive GAS (Supplementary Fig. 5c–e). Additionally, we observed that BAPTA-AM treatment suppressed TBK1 phosphorylation in response to GAS infection in a dose-dependent manner (Fig. 4e). These results suggested Ca$^{2+}$ as a key signaling molecule necessary for TBC1D9 recruitment, TBK1 activation, and autophagosome formation during GAS infection.

Because elevated Ca$^{2+}$ levels during GAS infection were dependent upon the pore-forming toxin SLO, we examined whether extracellular Ca$^{2+}$ is responsible for TBC1D9-mediated TBK1 activation. Treatment with extracellular BAPTA had an insignificant effect on TBK1 activation during GAS infection (Supplementary Fig. 5f, g). Dibromo-BAPTA-AM, which is the low affinity Ca$^{2+}$ chelator BAPTA-AM variant, was less effective in suppressing TBK1 activation than BAPTA-AM (Supplementary Fig. 5f, g). This result corresponds with affinity differences between BAPTA-AM ($K_d$ of Ca$^{2+}$-binding: 0.59 μM and dibromo-BAPTA-AM: 1.6 μM, no Mg$^{2+}$)[38]. Although BAPTA-AM treatment inhibited TBK1 activation, a slower Ca$^{2+}$ chelator (EGTA-AM) did not (Supplementary Fig. 5f, g), suggesting that GAS invasion rapidly triggers Ca$^{2+}$ signaling. To determine whether Ca$^{2+}$ from the ER is involved in TBK1 activation, we used the inositol trisphosphate receptor (IP3R) antagonists (xestospongin C (XeC), araguspongine B (ArB), and 2-aminoethoxydiphenylborane (2-APB)); the ryanodine receptor antagonist (dantrolene); and SERCA inhibitor (thapsigardin). Dantrolene and thapsigardin treatment did not affect TBK1 phosphorylation, but XeC, ArB, and 2-APB treatment impaired TBK1 phosphorylation during GAS infection in a dose-dependent manner (Supplementary Fig. 5f, g), suggesting induction of TBK1 activation through Ca$^{2+}$ transport via IP3R. We also found that U73122, a phospholipase C (PLC) antagonist, inhibited TBK1 activation, indicating that PLC-mediated IP3 signaling is involved in TBK1 activation during GAS infection.

As BAPTA-AM can affect the Na/K ATPase[39,40], we validated the Ca$^{2+}$ elevation in response to GAS infection upon treatment with BAPTA-AM. The Fura 2-AM ratio at 4 h after infection was suppressed by BAPTA-AM (Supplementary Fig. 5h). Moreover, BAPTA as well as U73122 inhibited Ca$^{2+}$ elevation during GAS infection (Supplementary Fig. 5h), suggesting that the increase in Ca$^{2+}$ is related to TBK1 activation. We further confirmed that the increase in cytosolic Ca$^{2+}$ during infection is not affected by knockout of STING, TBK1, and TBC1D9 (Supplementary Fig. 5i), suggesting that TBK1 activation is not required for Ca$^{2+}$ elevation.

To verify whether cytosolic Ca$^{2+}$ is required for the recruitment of TBC1D9 to bacteria and TBK1 activation, we overexpressed the Ca$^{2+}$ buffering proteins (PVALB and CALB). Overexpression of PVALB or CALB could block the TBC1D9 recruitment to intracellular GAS and TBK1 activation (Supplementary Fig. 6a, b).

To examine IP3R involvement in Ca$^{2+}$ signaling and TBC1D9-mediated TBK1 activation in xenophagy, we knocked down *IP3R* expression using siRNA in HeLa cells. Since IP3R has three isoforms (IP3R1, IP3R2, and IP3R3), we selectively knocked down their expression (Supplementary Fig. 7a). As shown in Fig. 4f, although TBK1 activation was decreased in all of four IP3Rs-knockdown cells compared to control cells, *IP3R1*-knockdown cells showed the most profound inhibition of TBK1 phosphorylation. If IP3R1 and TBC1D9 are required for TBK1 activation in different pathways, inhibition of two of these processes shows a larger inhibitory effect on TBK1 activation. However, the effects of single knockdown of *TBC1D9* and double knockdown of *TBC1D9* and *IP3R1* were comparable (Supplementary Fig. 7a–c). Moreover, *IP3Rs*-knockdown inhibited p62 recruitment to GAS and autophagosome formation (Supplementary Fig. 7d, e; Fig. 4i, j). We also found that depletion of IP3Rs significantly increased bacterial survival (Fig. 4k). These findings therefore suggested that IP3R-mediated elevations in Ca$^{2+}$ levels were involved in TBK1 activation and xenophagy against GAS infection.

**TBC1D9 is unnecessary for TBK1 activation in lysophagy.** Because elevations in Ca$^{2+}$ levels and TBC1D9-mediated TBK1 activation were induced in response to SLO following GAS infection, we hypothesized that Ca$^{2+}$ mobilization and TBC1D9 might play a role in TBK1 activation in response to endomembrane damage and as common regulators of lysophagy. To test this hypothesis, we treated cells with the lysosomal-damaging agent Leu-Leu-O-Me (LLOMe), which is converted to its membranolytic form through cathepsin D within lysosomes[41]. We observed increased in p-TBK1 in LLOMe-treated cells, and that this increase was not suppressed by BAPTA-AM treatment (Supplementary Fig. 8a), suggesting that intracellular Ca$^{2+}$ is not required for TBK1 activation by LLOMe. To monitor lysosomal-membrane damage and autophagosome formation against damaged lysosomes, we expressed galectin-3 as a marker[42]. Subsequent BAPTA-AM treatment did not affect the recruitment of either p62 or LC3 to galectin-3 puncta (Supplementary Fig. 8b–e). Additionally, LLOMe-induced LC3 recruitment to galectin-3 puncta was unchanged following *TBC1D9* KO (Supplementary Fig. 8f, g). These results suggested that Ca$^{2+}$ and TBC1D9 were not required for TBK1 activation and autophagosome formation during lysophagy.

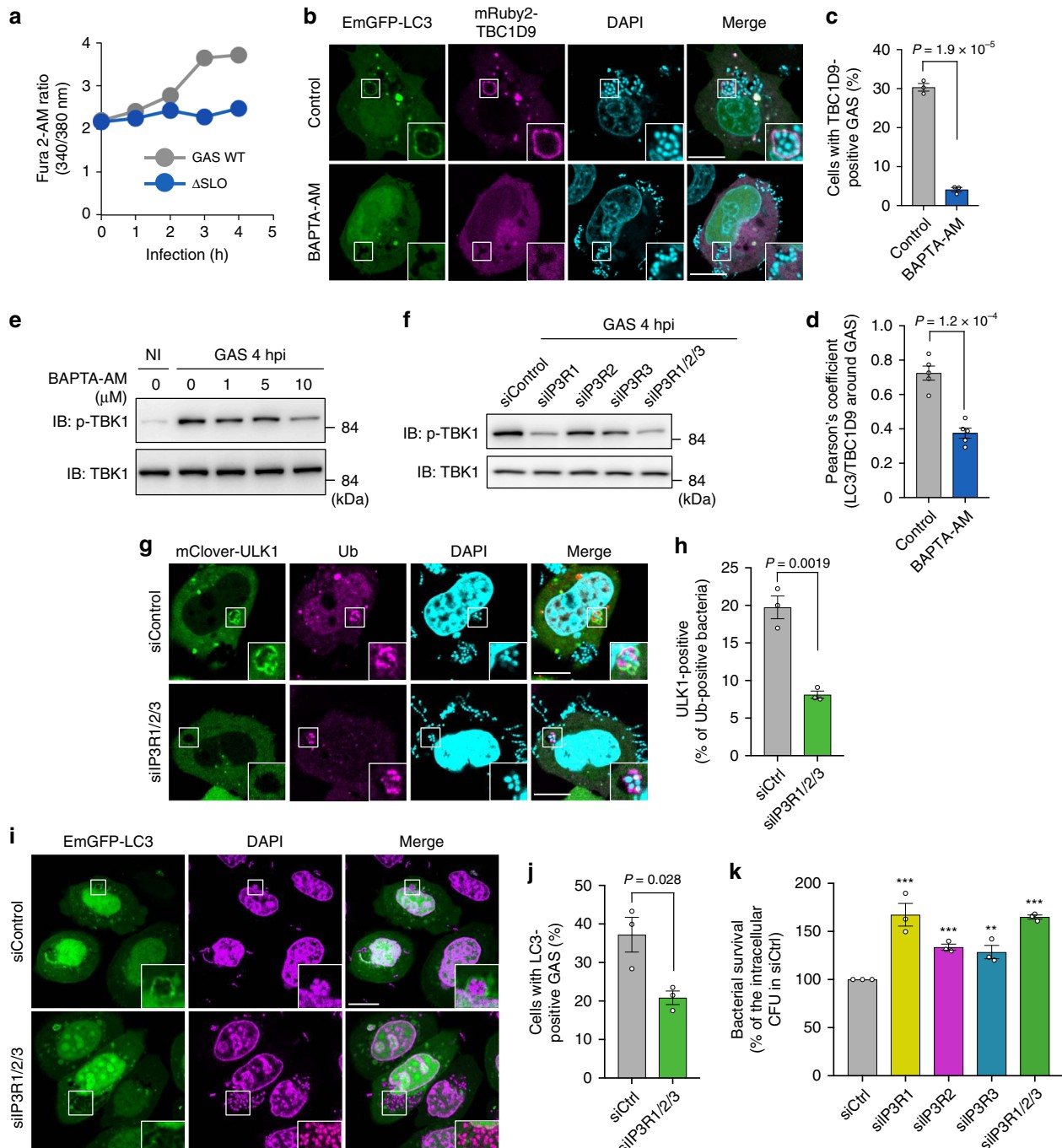

**Fig. 4 IP3R-mediated elevation in Ca²⁺ levels is required for TBC1D9 recruitment to bacteria and TBK1 activation. a** Representative trace of Fura 2-AM ratio during GAS WT or Δ*slo* infection. **b–d** HeLa cells expressing EmGFP-LC3 and mRuby2-TBC1D9 and treated with or without BAPTA-AM (10 μM) were infected with GAS for 4 h. **c** Confocal images of TBC1D9 recruitment during infection and **d** quantification of cells harboring TBC1D9-positive bacteria, and **d** quantification of colocalization between LC3 and TBC1D9. **e** Effects of BAPTA-AM treatment on TBK1 phosphorylation during GAS infection. **f** Effects of IP3Rs depletion on TBK1 phosphorylation during GAS infection. **g, h** Recruitment of ULK1 to intracellular GAS in IP3Rs-KD cells. HeLa cells transfected with mClover-ULK1 and the indicated siRNAs were infected with GAS, fixed at 4 h, and stained for endogenous ubiquitin. **g** Representative confocal images and **h** the frequency of ULK1-positive bacteria. **i, j** HeLa cells transfected with EmGFP-LC3 and the indicated siRNAs were infected with GAS for 4 h. **i** Representative confocal images and **j** the frequency of cells with LC3-positive bacteria. **k** HeLa cells transfected with the indicated siRNAs were infected with GAS and intracellular bacterial CFU was determined at 6 h p.i. Data in (**c**) (*n* > 200 cells per condition), (**d**) (*n* > 5 images per condition), (**h**) (*n* > 50 bacteria per condition), (**j**) (*n* > 200 cells per condition), and (**k**) represent the mean ± SEM of three independent experiments. *P* values calculated by two-tailed Student's *t* test. \*\**P* value < 0.01, \*\*\**P* value < 0.001.

**TBC1D9 and Ca²⁺ are required in mitophagy**. IP3R-mediated $Ca^{2+}$ levels are reportedly involved in mitophagy[43]. Therefore, we determined whether $Ca^{2+}$ and TBC1D9 are involved in TBK1 activation during mitophagy. We observed endogenous TBC1D9 localization in mCherry-Parkin-expressing HeLa cells exposed to antimycin and oligomycin (AO), and mCherry-Parkin-expressing cells showed partial localization of TBC1D9 to mitochondria in control cells and perinuclear mitochondrial clustering and endogenous TBC1D9 colocalization with the clustered mitochondria in AO-treated cells (Fig. 5a). Additionally, EmGFP-TBC1D9 localized with clustered mitochondria only in Parkin-expressing cells (Fig. 5b, c), suggesting TBC1D9 recruitment to depolarized mitochondria through a Parkin-dependent mechanism. Moreover, we found that TBC1D9 lacking a $Ca^{2+}$-binding motif (EF) and UBR failed to be recruited to clustered mitochondria in AO-treated HeLa cells (Fig. 5d). Furthermore, TBC1D9 recruitment to depolarized mitochondria was abolished by BAPTA-AM treatment (Fig. 5e). These results suggested that TBC1D9 was recruited to damaged mitochondria via $Ca^{2+}$ signaling and a Parkin-dependent ubiquitination mechanism.

**TBC1D9 is required for TBK1 activation in mitophagy**. We next examined whether TBC1D9 and $Ca^{2+}$ are involved in TBK1-mediated mitophagy. We found that *TBC1D9*-KO decreased TBK1 phosphorylation triggered in response to AO treatment (Fig. 6a). To investigate the efficiency of recruitment of the autophagy adaptor NDP52 to depolarized mitochondria, we analyzed NDP52 localization and colocalization with the mitochondrial import-receptor subunit TOM20 in AO-treated cells. As expected, a significantly smaller proportion of TOM20 signals were GFP-NDP52-positive in *TBC1D9*-KO cells, as well as *TBK1*-KO cells (Fig. 6b, c). These results suggested that TBC1D9 was required for TBK1 activation to recruit NDP52 to depolarized mitochondria.

TBC1D9-mediated TBK1 activation and NDP52 recruitment lead to recruitment of the ULK1 complex to invading bacteria in xenophagy (Fig. 2e, f), and ULK1 recruitment to depolarized mitochondria also involves TBK1 and NDP52 in mitophagy[14]. We then examined the localization of ULK1 complex during mitophagy in *TBC1D9*-KO cells. Although mClover-ATG13 significantly colocalized with Parkin-positive clustered mitochondria in AO-treated WT cells, colocalization was limited in partial fragments of Parkin-positive mitochondria in *TBC1D9*-KO cells (Fig. 6d, e), suggesting that TBC1D9 was involved in recruitment of the ULK1 complex to depolarized mitochondria in mitophagy. We then analyzed mitophagy in *TBC1D9*-KO cells by measuring the degradation of cytochrome C oxidase subunit II (COXII), a mitochondrial DNA-encoded protein after AO treatment. After 24-h AO treatment, the remaining COXII was significantly increased in *TBC1D9*-KO cells compared to that in WT cells (Fig. 6f, g), suggesting that efficient mitophagy activity requires TBC1D9.

We further examined the effects of BAPTA-AM treatment on AO-induced TBK1 phosphorylation. We found that BAPTA-AM treatment reduced TBK1 phosphorylation in a dose-dependent manner (Fig. 6h), and that NDP52 recruitment to depolarized mitochondria in response to AO treatment was also reduced by BAPTA-AM treatment (Fig. 6i, j), indicating that TBK1-mediated NDP52 recruitment to damaged mitochondria requires intracellular $Ca^{2+}$ signaling.

## Discussion

TBK1 is a key signaling molecule in cells that promotes the response to microbial pathogens by activating IRF3 to stimulate IFN production and recruitment of autophagy adaptors for autophagy regulation[32]. Here, we demonstrated that TBK1 activation during xenophagy and mitophagy required $Ca^{2+}$ signaling, and that this process was mediated by TBC1D9, a novel regulator of autophagy. We found that elevations in cytosolic $Ca^{2+}$ levels in response to GAS invasion triggered TBC1D9 recruitment to GAS and ubiquitin binding. Additionally, TBC1D9 interacted with activated TBK1 to promote TBK1 self-association and further activity. Accordingly, $Ca^{2+}$-mediated TBC1D9 and TBK1 promoted xenophagy through NDP52 and ULK1 complex recruitment (Fig. 7). Furthermore, we found that TBC1D9-regulated TBK1 activation and recruitment of NDP52 and ULK1 complex to damaged mitochondria (Fig. 7). The role of TBC1D9 as a Rab regulator is not identified, but we have revealed a function of TBC1D9 through the binding ability to $Ca^{2+}$, which is unique in RabGAP family proteins.

We unexpectedly found that TBK1 activation during GAS infection involved neither STING nor IRF activation (Supplementary Fig. 1). Additionally, TBK1 activation during mitophagy occurred independent of IRF3 phosphorylation. In response to microbial infection, a variety of pattern recognition receptors, such as cytosolic DNA/RNA sensors, and endosomal Toll-like receptor [3,7,9], could activate TBK1 to induce IRF3 and NF-kB signaling[44–47]. Since TBK1 serves as important components of multiple signaling pathway, TBK1 needs to specifically activate discrete signaling complex for different cellular responses. Upon stimulations, TBK1 is recruited to signaling complex via its C-terminal adaptor domain, and local clustering of TBK1 enables TBK1 to dimerize for trans-autophosphorylation[48]. Therefore, spatial regulation of TBK1 via unique binding partner and amplification of TBK1 activation are suggested to be important for TBK1 substrate specificity[48]. During DNA-sensing pathway, translocated STING to the Golgi defines TBK1 activation to phosphorylate IRF3 (refs. [21,29]), whereas phosphorylated TBK1 was observed in proximal to invading GAS (Fig. 1c). Given that calcium chelator and knockout of *TBC1D9* suppressed TBK1 activation and recruitment of TBK1 to ubiquitin-tagged GAS (Figs. 1a–d, 4e), calcium and TBC1D9 would be spatial or amplifying regulators of TBK1 during selective autophagy. Since SINTBAD and NAP1 were also reported to be involved in selective autophagy against *Salmonella Typhimurium*[8], further study of the association between these regulators and TBC1D9 might clarify TBK1 activation mechanism during selective autophagy against bacteria.

We identified TBC1D9 as a novel regulator of TBK1-mediated xenophagy and mitophagy. Although we screened TBC/RabGAP family proteins, negative regulators or Rab GTPase, GAP-inactive form of TBC1D9 also could regulate TBK1 activation pathway (Fig. 1c, d), indicating that function of TBC1D9 in TBK1 activation during autophagy is not mediated by substrate Rab GTPase. TBC1D9 is the only EF-hand motif-containing protein in RabGAP family, and substrate Rab proteins of TBC1D9 has not identified yet. In response to GAS infection, TBC1D9 appeared to promote TBK1 activation, given that TBK1 activation was diminished in *TBC1D9*-KO cells. Based on our observation that TBC1D9 interacted with p-TBK1, it is possible that TBC1D9 might play a role as an adaptor or scaffold to promote TBK1 trans-autophosphorylation at the target site. Because tripartite motif-containing protein 23 (TRIM23) is involved in the TBK1–p62 axis through its promotion of TBK1 self-association during virus-induced autophagy[49], it would be instructive to determine whether $Ca^{2+}$ signaling is also involved in TRIM23-mediated TBK1 activation, and whether TBC1D9 cooperates with TRIM23 to promote xenophagy. In TBK1 activation mechanism, not only dimerization but conformation changes in TBK1 have been also indicated to control its activation[50], and Raf kinase inhibitory protein (RKIP) is reported to be critical for TBK1

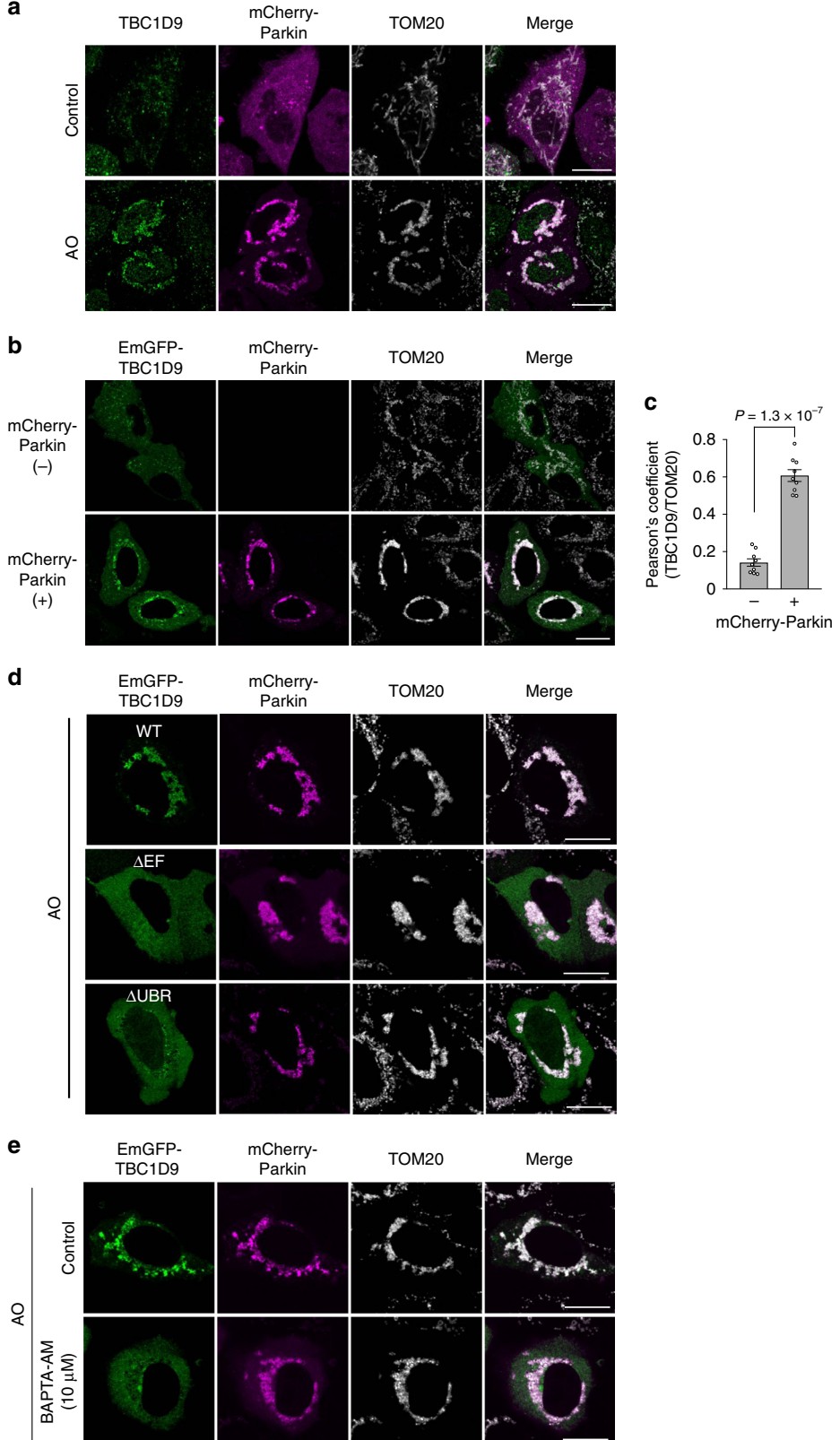

**Fig. 5 Parkin-dependent recruitment of TBC1D9 to depolarized mitochondria involves Ca²⁺ signaling. a** HeLa cells expressing mCherry-Parkin were treated with AO for 4 h and stained for TBC1D9 and TOM20. Scale bars, 10 μm. **b, c** HeLa cells expressing EmGFP-TBC1D9 and mCherry-Parkin were treated with AO for 4 h and stained for TOM20. **b** Representative confocal images and **c** Pearson's coefficient between TBC1D9 and TOM20. Data in (**c**) represent the mean ± SEM of three independent experiments (*n* > 5 images per condition). **d** HeLa cells expressing mCherry-Parkin and EmGFP-TBC1D9 (WT, ΔEF-hand motif, and ΔUBR) were treated with AO for 4 h and stained for TOM20. **e** HeLa cells expressing EmGFP-TBC1D9 and mCherry-Parkin were treated with AO and BAPTA-AM for 4 h and stained for TOM20. *P* values calculated by two-tailed Student's *t* test.

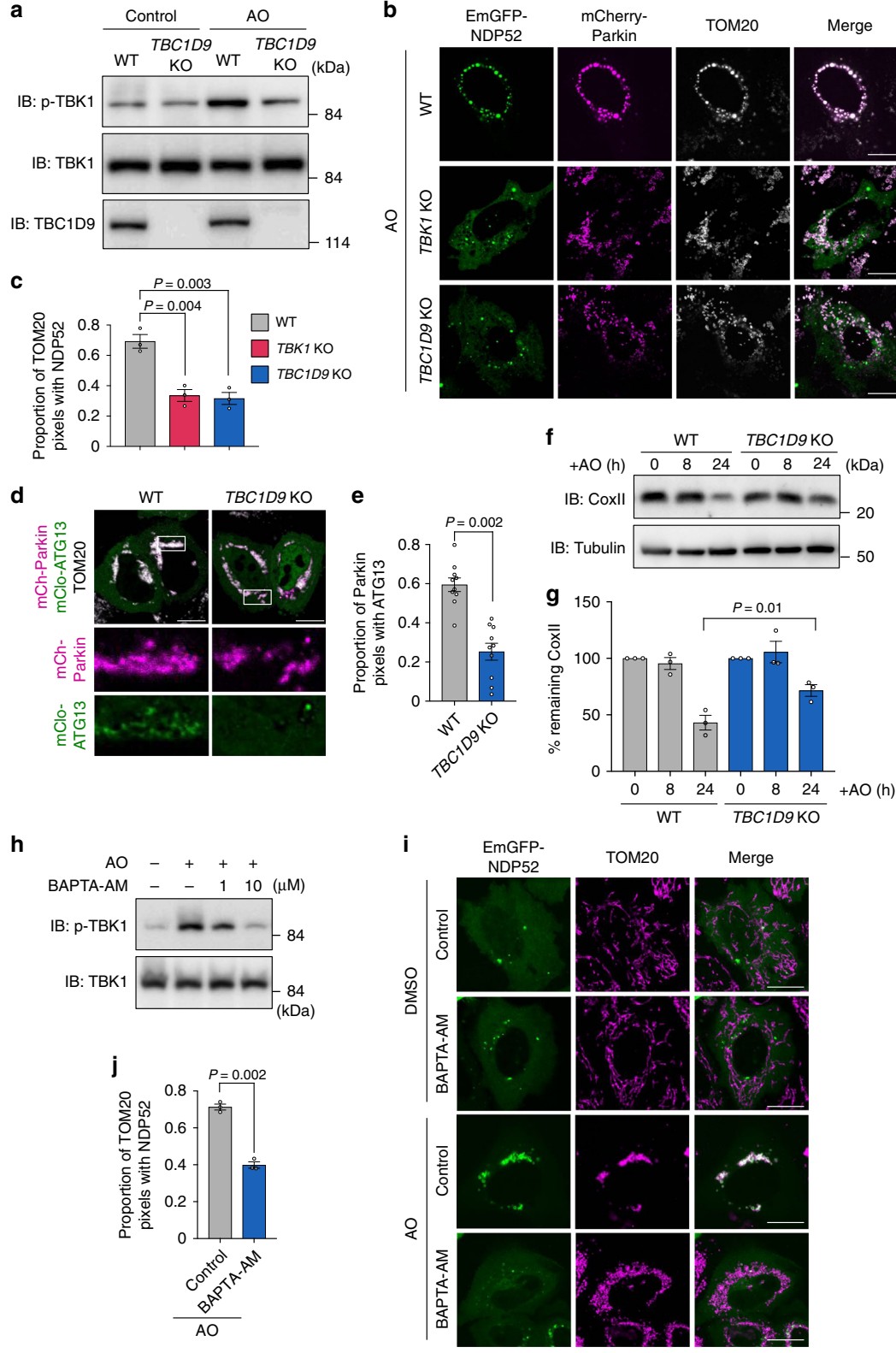

activation during virus infection maybe through inducing conformational change of TBK1 (ref. [51]). Although TBC1D9 deficiency diminished TBK1 self-association, this does not exclude the possibility that TBC1D9 is also involved in the conformational change to regulate TBK1 activation.

TBK1 and NDP52 mutually regulate their recruitment to cytosolic bacteria[9,15], and associate with several autophagy regulators such as LC3 (ref. [52]), WIPI2 (ref. [53]), and the ULK1 complex[15]. Since we demonstrated that TBC1D9 is required for the recruitment of TBK1, NDP52, ULK1, ATG13, and LC3 to invading GAS, TBC1D9-mediated TBK1 activation would trigger NDP52-ULK1 complex recruitment for xenophagy. In addition, since ULK1 directs ATG16L1 to intracellular bacteria for xenophagy, TBC1D9-regulated TBK1 activation might

**Fig. 6 Ca$^{2+}$ and TBC1D9 are required for TBK1 activation during mitophagy. a** HeLa WT or *TBC1D9*-KO cells expressing mCherry-Parkin were depolarized with AO for 2 h. **b, c** HeLa WT, *TBK1*-KO, and *TBC1D9*-KO cells expressing EmGFP-NDP52 and mCherry-Parkin were depolarized with AO for 4 h and immunostained for TOM20. **b** Representative confocal micrographs and **c** the proportion of TOM20 puncta colocalized with NDP52 from at least 20 randomly selected cells in each experiment. **d, e** HeLa WT and *TBC1D9*-KO cells expressing mCherry-Parkin and mClover-ATG13 were depolarized with AO for 4 h and immunostained for TOM20. **d** Representative confocal images and **e** quantification of ATG13 recruitment to Parkin-positive clustered mitochondria. **f, g** Mitophagy activity in *TBC1D9*-KO cells. **f** Cell lysates from *TBC1D9*-KO cells expressing mCherry-Parkin after AO treatment were immunoblotted and **g** quantified for COXII degradation. **h** HeLa cells expressing mCherry-Parkin were depolarized with AO for 2 h in the presence or absence of BAPTA-AM. **i, j** HeLa cells expressing EmGFP-NDP52 and FLAG-Parkin were depolarized with AO for 4 h in the presence or absence of BAPTA-AM and immunostained for TOM20. **i** Representative confocal micrographs and **j** the proportion of TOM20 puncta colocalized with NDP52 from at least 20 randomly selected cells in each experiment. Data in (**c**) (*n* > 20 cells per condition), (**e**) (*n* > 5 cells per condition), (**g**), and (**j**) (*n* > 10 cells per condition) represent the mean ± SEM of three independent experiments. *P* values calculated by two-tailed Student's *t* test.

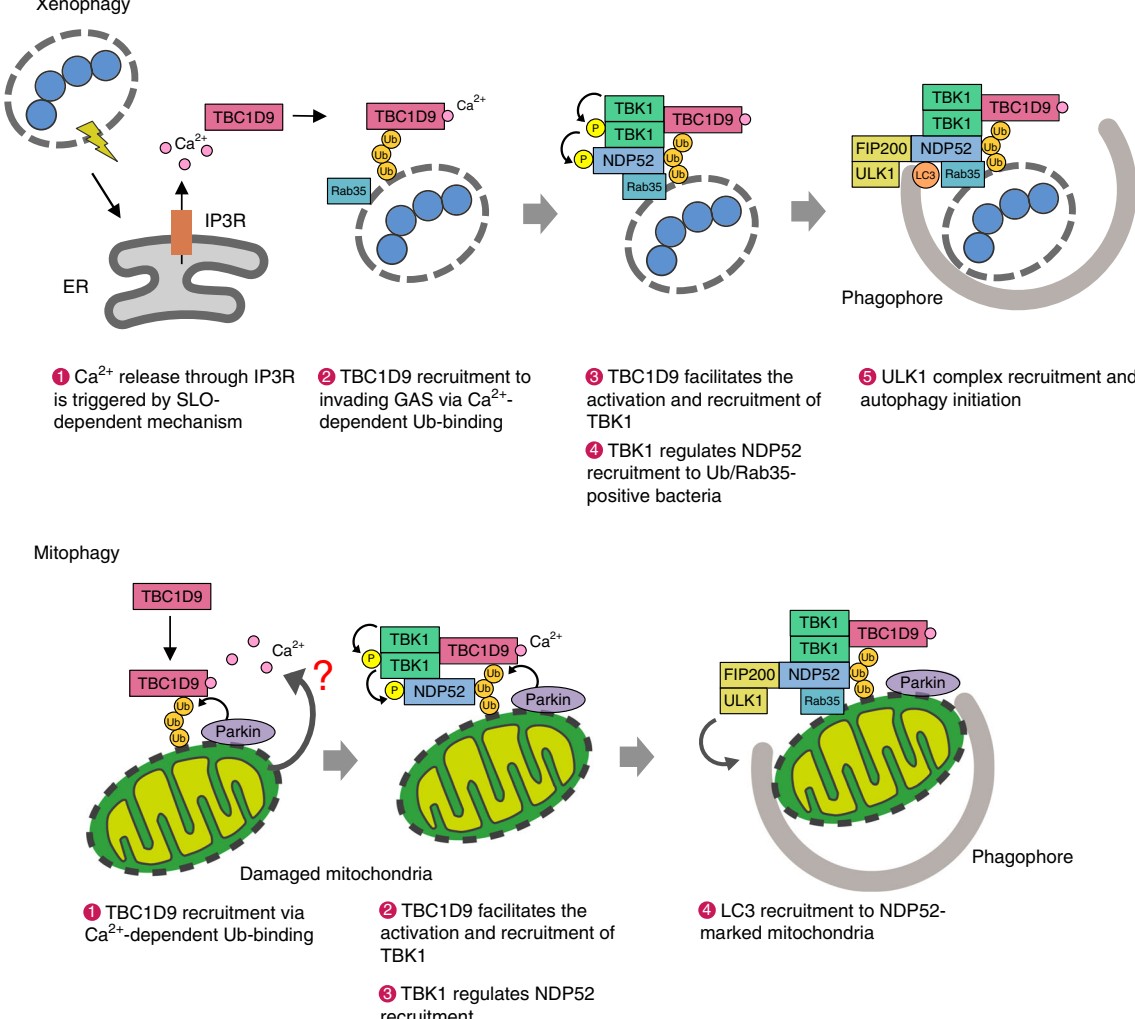

**Fig. 7 Model of Ca$^{2+}$-dependent TBC1D9 and TBK1 activation in xenophagy and mitophagy.** In xenophagy, TBC1D9 is activated by ER-derived Ca$^{2+}$ and recruits to ubiquitin (Ub). Then recruited TBC1D9 activates TBK1 and activated TBK1 regulates NDP52 recruitment to Ub. Finally, ULK1 complex is recruited by NDP52/Ub/RAB35 for initiation of autophagy. In mitophagy, TBC1D9 targets depolarized mitochondria in a Parkin-dependent manner and facilitates TBK1 activation on the mitochondria. TBK1 mediates NDP52 recruitment for mitophagy.

result in autophagosome formation through ULK1. Notably, although the ULK1 complex is critical for LC3-positive autophagosome formation during starvation, FIP200 is dispensable for LC3 recruitment to bacteria in xenophagy[54]. Our results in this study showed that depletion of ULK1 complex components including FIP200 resulted in increased bacterial survival, indicating the importance of ULK1 complex in the xenophagy of GAS in cells. A recent study revealed that

bacteria-triggered vacuolar damages induce V-ATPase-mediated ATG16L1 recruitment to the invading bacteria in xenophagy[54]. Moreover, NDP52 is believed to tether LC3C on the autophagic membrane in xenophagy[8,15,52], TBC1D9 can also bind to LC3[17], and TBK1 is reported to phosphorylate LC3 to control its dynamics. Therefore, the ULK1 complex might be involved in xenophagy processes besides the initiation of autophagosome formation. In fact, TBK1 is known to control

lysosome fusion and was recently reported to phosphorylate STX17 to control the assembly of ATG13 and FIP200 (refs. [10,55]). We also showed that STX17 colocalized with LC3 during GAS infection[56]. Therefore, further studies to reveal the TBK1-mediated ULK1 complex recruitment in xenophagy might shed light on a novel regulatory mechanism of xenophagy.

We found that $Ca^{2+}$ signaling promoted ubiquitin binding and TBC1D9 recruitment to ubiquitinated autophagy targets based on its localization with invading GAS dependent upon the TBC1D9 $Ca^{2+}$-binding motif and UBR. Moreover, this was confirmed by our observation of impaired TBC1D9 recruitment following BAPTA-AM treatment. Although we cannot exclude the possibility that TBC1D9 recognizes autophagy targets through interactions with molecules other than ubiquitin during GAS infection, our results showing Parkin-dependent TBC1D9 recruitment to depolarized mitochondria supports its likely targeting of ubiquitinated mitochondria during mitophagy.

We showed that IP3R-mediated elevations in $Ca^{2+}$ levels during GAS infection triggered TBK1 activation via TBC1D9. Additionally, we demonstrated that $Ca^{2+}$/TBC1D9 signaling was required for mitophagy but not for lysophagy. An important unanswered question concerns why lysophagy did not involve $Ca^{2+}$ and TBC1D9. Recruitment of NDP52 and TBK1 requires the membrane damage sensor galectin-8 during *S. Typhimurium* infection[57], and galectin-3 senses endosomal damages and mobilizes core autophagy regulators ATG16L1, ULK1, and Beclin 1 to the damaged endomembranes[58]. In contrast, galectin-3 negatively regulates xenophagy during GAS infection[59], and RAB35 but not galectin-8 is required for the recruitment of NDP52 and TBK1 against GAS infection[9]. These results imply that danger sensors largely differ among the stimuli and that various sensors utilize TBK1 kinase to initiate autophagy. Previous studies showed that STING-mediated TBK1 activation also involves intracellular $Ca^{2+}$ mobilization via IP3R, and small-interfering RNA assays revealed that IP3R and $Ca^{2+}$ are involved in mitophagy[43]. These findings and our results in the present study suggest that IP3R might be critical to $Ca^{2+}$/TBC1D9-mediated TBK1 activation.

Although we showed that elevation in $Ca^{2+}$ levels during GAS infection is dependent upon the cholesterol-dependent cytolysin (CDS) SLO and IP3R, we were unable to elucidate the mechanism underlying SLO-mediated IP3R activation. Since a previous study showed that listeriolysin O (LLO), a CDS from *Listeria monocytogenes*, induces increased $Ca^{2+}$ levels, which activate the phospholipase C–IP3R pathway[60], SLO also might share similar characteristics with LLO. GAS secretes various effector proteins into the host cytosol via SLO; therefore, there is a possibility that $Ca^{2+}$ mobilization via IP3R is a consequence of any number of SLO-secreted molecules. Further studies focused on GAS-mediated induction of $Ca^{2+}$ signaling are needed to clarify the roles of $Ca^{2+}$ signaling in autophagy, as well as innate and adaptive immunity. Moreover, how TBC1D9 and TBK1 can target ubiquitin-positive bacteria through $Ca^{2+}$ signaling is also unclear. In response to dsDNA, the autophagy regulator ATG9A-mediated membrane traffic regulates the assembly of STING and TBK1 to activate innate immune responses[61]. Considering the fact that TBC1D9 is a member of membrane traffic regulators, a membrane-mediated regulatory system for TBK1 activation might exist in selective autophagy.

GST pulldown and PLA assays showed that TBC1D9 preferentially interacted with Lys63-linked polyubiquitin. TBC1D9 reportedly interacts with LC3 (ref. [17]), and because a unifying property of ubiquitin-binding autophagy receptors is their ability to bind to both ubiquitin and LC3, TBC1D9 might play a role as an autophagy receptor. However, given our result showing that TBC1D9 did not decrease during GAS infection (Fig. 1a), TBC1D9 would likely be a specific regulator in response to $Ca^{2+}$ signaling rather than a master regulator of autophagy.

In conclusion, our data highlighted the functional significance of TBC1D9 in regulating TBK1 activation associated with selective autophagy. Since dysregulation of TBK1 activity is often associated with not only infectious disease, but also autoimmune disease and cancer, our identification of $Ca^{2+}$-dependent TBC1D9-mediated TBK1 activation might help elucidate the pathogenesis of various diseases related to TBK1 and autophagy, as well as identify therapeutic targets.

## Methods

**Cell culture and transfection**. HeLa cells were purchased from American Type Culture Collection (ATCC) and maintained in 5% $CO_2$ at 37 °C in Dulbecco's modified Eagle medium (Nacalai Tesque) supplemented with 10% fetal bovine serum albumin (BSA; Gibco; Thermo Fisher Scientific) and 50 μg mL$^{-1}$ gentamicin (Nacalai Tesque). Polyethylenimine (Polysciences) and Lipofectamine 3000 (Invitrogen) were used for transfection.

**Reagents**. A23187 (Tocris Bioscience), dantrolene (Tocris Bioscience), 2-APB (Tocris Bioscience), BAPTA-AM (Dojindo), BAPTA (Dojindo), EGTA (Dojindo), EGTA-AM (AAT Bioquest), 5,5′-difluoro-BAPTA-AM (PromoCell Gmbh), xestospongin C (Abcam), araguspongin B (Cayman Chemical), U73122 (Aobious, Gloucester), thapsigardin (Nacalai Tesque), brefeldin A (Nacalai Tesque), L-Leucyl-L-Leucine methyl ester (Cayman Chemical), antimycin A (Santa Cruz Biotechnology), and oligomycin (Calbiochem) were purchased.

**Plasmids and siRNA transfection**. Human ULK1, ATG13, ubiquitin, IFN-stimulated gene 15 (ISG15), ubiquitin-fold modifier 1 (UFM1), ubiquitin-related modifier 1 (URM1), parvalbumin (PVALB), and calbindin 1 (CALB) were amplified by polymerase chain reaction (PCR) from total mRNA derived from HeLa, KYSE, and HEK293T cells pcDNA-6.2/N-EmGFP-DEST, pcDNA-6.2/N-3xFLAG-DEST, pcDNA-6.2/N-mCherry-DEST, pGEX-6P-1-DEST, and pLenti6/V5-DEST using Gateway technology (Invitrogen). pcDNA-6.2/N-mClover-DEST (N-terminal tagged) and pcDNA-6.2/N-mRuby2-DEST (N-terminal tagged) vectors were made by replacement from pcDNA-6.2/N-EmGFP to the mClover or mRuby2 gene fragment amplified from pMK289 (Addgene, #72827) and mRuby2-C1 (Addgene, #54768), respectively. Human parvalbumin (PVALB) and calbindin 1 (CALB) were amplified from cDNA clones (100000505 and 4592497; DNA-FORM). G-CaMP3 (Addgene, #22692) was obtained from Addgene. The resulting constructs were fused at their respective N-terminus with the corresponding tags. TBC1D9 and TBK1 were mutated by site-directed mutagenesis using a PrimeSTAR mutagenesis basal kit (Takara Bio). The following primers were used in this study: *ULK1* fw: 5′-GGAACCAATTCAGTCGACTGGATGGAGCCCGGCCGCG-3′, rev: 5′-GAAAGCTGGGTCTAGATATCTCAGGCACAGATGCCAGTC-3′; *ATG13* fw: 5′-GGAACCAATTCAGTCGACTGGATGGAAACTGATCTCAATT-3′, rev: 5′-GAAAGCTGGGTCTAGATATCTTACTGCAGGGTTTCCACA-3′; *ISG15* fw: 5′-GGAACCAATTCAGTCGACTGGATGGGCTGGGACCTGACGG-3′, rev: 5′-GAAAGCTGGGTCTAGATATCTTAGCTCCGCCCGCCAGGC-3′; *UFM1* fw: 5′-GGAACCAATTCAGTCGACTGGATGTCGAAGGTTTCCTTTA-3′, rev: 5′-GAAAGCTGGGTCTAGATATCTTAACAACTTCCAACACGA-3′; *URM1* fw: 5′-GGAACCAATTCAGTCGACTGGATGGCTGCGCCCTTGTCAG-3′, rev: 5′-GAAAGCTGGGTCTAGATATCTCAGGATGGAGGAGTACTC-3′; *PVALB* fw: 5′-GGAACCAATTCAGTCGACTGGATGTCGATGACAGACTTG-3′, rev: 5′-GAAAGCTGGGTCTAGATATCTTAGCTTTCAGCCACCAGA-3′; *CALB* fw: 5′-GGAACCAATTCAGTCGACTGGATGGCAGAATCCCACCT-3′, rev: 5′-GAAAGCTGGGTCTAGATATCCTAGTTATCCCCAGCACA-3′.

For knockdown experiments, cells were transfected with TBC1D9 siRNA oligonucleotides (s23161; Thermo Fisher Scientific), FIP200 siRNA oligonucleotides (HSS19064; Thermo Fisher Scientific), ATG13 siRNA oligonucleotides (5′-CCAUGUGUGUGGAGAUUUCACUUAA; Thermo Fisher Scientific), ULK1 siRNA oligonucleotides (s15964, Thermo Fisher Scientific), ULK2 siRNA oligonucleotides (s18705; Thermo Fisher Scientific), IP3R1 siRNA oligonucleotides (s7632; Thermo Fisher Scientific), IP3R2 siRNA oligonucleotides (s7634; Thermo Fisher Scientific), IP3R3 siRNA oligonucleotides (s265; Thermo Fisher Scientific), or nontargeting siRNAs (12935300, Thermo Fisher Scientific) using Lipofectamine 3000 (Invitrogen). Knockdown was confirmed by immunoblotting.

**Antibodies and western blotting**. For western blotting, the following antibodies were used: TBK1 (EP611Y; ab40676, Abcam, 1:1000), p-TBK1 (Ser172) (D52C2; 5483; Cell Signaling Technology, 1:1000), STING/TMEM173 (19851-1-AP; Proteintech, 1:1000), p-IRF3 (Ser396) (4D4G; 4947; Cell Signaling Technology, 1:1000), IRF3 (SL-12; sc-33641, Santa Cruz Biotechnology, 1:500), TBC1D9 (A301-028A; Bethyl Laboratories, 1:1000), IP3R (D53A5; 3763; Cell Signaling Technology, 1:1000), ULK1 (D9D7; 6439; Cell Signaling Technology, 1:1000), ULK2 (C2C3;

GTX111476; GeneTex, 1:500), FIP200 (17250-1-AP; Proteintech, 1:1000), ATG13 (E1Y9V; 13468; Cell Signaling Technology, 1:1000), LC3B (ab51520; Abcam, 1:1000), β-actin (D6A8; 8457; Cell Signaling Technology, 1:1000), IP3R2 (A-5; sc-3988434; Santa Cruz Biotechnology, 1:500), FLAG (M2; A2220; Sigma-Aldrich, 1:1000), TOM20 (F-10; sc-17764; Santa Cruz Biotechnology, 1:2000), GAPDH (6C5; sc-32233; Santa Cruz Biotechnology, 1:10000), COXII (12C4F12; ab110258; Abcam, 1:500), GFP (GF200; 04363-24; Nacalai Tesque, 1:1000). Horseradish peroxidase-conjugated anti-rabbit IgG (Jackson Laboratories), anti-mouse IgG (Jackson Laboratories) and anti-rabbit IgG (conformation-specific; L27A9; 5127; Cell Signaling Technology) were used as secondary antibodies for immunoblots.

**Generation of KO cell lines by CRISPR/Cas9.** CRISPR/Cas9 was used to KO *STING*, *TBC1D9*, and *TBK1*. CRISPR guide (g)RNAs were designed to target an exon common to all splicing variants of the gene of interest (5′-GAGAGTGTG CTCTGGTGGC-3′ for *STING*, 5′-AACCCGGAGGAGGTGTTGC-3′ for *TBC1D9*, and 5′-GAGCACTTCTAATCATCTG-3′ for *TBK1*). HeLa cells were transfected with the vector hCAS9 (Addgene #41815) and a gRNA-hyg vector containing the CRISPR target sequence. Untransfected cells were removed by selection on plates containing 300 μg mL$^{-1}$ hygromycin B (Nacalai Tesque) and 750 μg mL$^{-1}$ geneticin (G418; Nacalai Tesque). Single colonies were expanded, and depletion of the target gene was confirmed by immunoblot. As a secondary screen for some KO lines, genomic DNA was isolated, and target regions were amplified by PCR and sequenced to confirm the presence of the desired frameshift insertions and deletions.

**Bacterial infection.** GAS strain JRS4 (M6$^+$F1$^+$) was grown in Todd–Hewitt broth (BD Diagnostic Systems) supplemented with 0.2% yeast extract. Cells were infected with GAS, as described previously[25]. Briefly, cell cultures in media without antibiotics were infected for 1 h at a multiplicity of infection of 100. Infected cells were washed with phosphate-buffered saline (PBS) and treated with 100 μg mL$^{-1}$ gentamicin for an appropriate period in order to kill bacteria that were not internalized.

**Bacterial viability assay.** HeLa cells ($5 \times 10^4$ cells/well) were cultured in 24-well culture plates and infected as described under "Bacterial infection". After an appropriate incubation period, cells were lysed in sterile distilled water, and then serial dilutions of the lysates were plated on THY agar plates. Colony counting was used to determine the numbers of invaded and surviving GAS; the bacterial survival data are presented as the ratio of "intracellular live GAS at 6 h" to "total intracellular GAS at 2 h".

**Immunoprecipitation.** Cells were harvested, washed with PBS, and lysed for 30 min on ice in lysis buffer containing 10 mM Tris-HCl (pH 7.4), 150 mM NaCl, 10 mM MgCl$_2$, 1 mM ethylenediaminetetraacetic acid (EDTA), 1 % Triton X-100, and a proteinase-inhibitor cocktail (Nacalai Tesque). Lysates were then centrifuged, and supernatants were precleared for 30 min at 4 °C with a Protein G Sepharose Fast Flow column (GE Healthcare Life Sciences). After a brief centrifugation, supernatants were reacted for 2 h at 4 °C with the appropriate antibodies and incubated with Protein G Sepharose beads (GE Healthcare Life Sciences) while shaking for another 1 h at 4 °C. Rabbit immunoglobulin fraction (X0903; Dako; Agilent Technologies, Santa Clara, CA, USA) was used as a negative control for immunoprecipitation. Immunoprecipitates were collected by brief centrifugation, washed five times with lysis buffer, and analyzed by immunoblot, as described previously.

For the binding experiment shown in Fig. 1i, lysate of HeLa WT or *TBC1D9*-KO cells transfected with GFP-TBK1 S172E and FLAG-TBK1 S172E were immunoprecipitated with GFP-Trap (ChromoTek).

**Protein expression and purification.** GST-fusion proteins constructed in pGEX-6P-1 (GE Healthcare Life Sciences) were transformed into *Escherichia coli* BL21 (DE3) cells, which were then cultured at 37 °C in Luria–Bertani medium supplemented with 100 μg mL$^{-1}$ ampicillin and induced for 3 h at 37 °C with 0.3 mM isopropyl β-D-thiogalactopyranoside (Nacalai Tesque). Cells were harvested by centrifugation, washed with PBS, lysed in 40 mM Tris-HCl (pH 7.5), 5 mM EDTA, and 0.5 % Triton X-100, sonicated, and cleared by centrifugation. The resulting supernatant was incubated with Glutathione Sepharose 4 Fast Flow beads (GE Healthcare Life Sciences) for 2 h at 4 °C. After several washes with buffer, the beads were used directly in pulldown assays.

**Pulldown assay.** HeLa cells were transfected with expression constructs encoding the protein of interest. After 24 h, cells were lysed in 50 mM HEPES (pH 7.4), 250 mM NaCl, 10 mM MgCl$_2$, 1 % Triton X-100, and a proteinase-inhibitor cocktail (Nacalai Tesque). Lysates were incubated for 2 h at 4 °C with beads precoated with GST-fusion proteins, and GST-fusion proteins. Beads were washed five times with buffer and analyzed by immunoblot.

For the binding experiment shown in Fig. 3j, beads precoated with GST-fusion proteins were used to pulldown 230 nM Lys48- or Lys63-tetra ubiquitin for 1 h at 4 °C. Beads were washed five times with lysis buffer and analyzed by immunoblot using antiubiquitin antibody (P4D1; 14049; Cell Signaling Technology).

**In situ PLA.** The PLA was performed using Duolink (Olink Bioscience). HeLa cells grown on coverslips were infected with or without GAS for 4 h, fixed in 4% paraformaldehyde (PFA) for 15 min, washed, permeabilized with 0.1% Triton X-100, and blocked with 2% BSA blocking buffer for 1 h at room temperature. Cells were probed overnight at 4 °C with primary antibodies diluted in 2% BSA blocking buffer. As a negative control, cells were incubated in antibody diluent with either primary antibodies. Labeling with secondary antibodies, ligation, and signal amplification were performed according to the manufacturer's recommendations. PLA dots were imaged with an FV1000 confocal microscope (Olympus).

**Fluorescence microscopy.** For immunofluorescence, the following antibodies were used: p-TBK1 (Ser172) (D52C2; 5483; Cell Signaling Technology, 1:100), p62 (D-3; sc-28359; Santa Cruz Biotechnology, 1:100), polyubiquitin (FK2; MFK-004; Nippon Bio-Test Laboratories, 1:200), p62 (H-290; sc-25575; Santa Cruz Biotechnology,1:100), TBC1D9 (A301-028A; Bethyl Laboratories, 1:100), Lys63-specific ubiquitin (Apu3; 05-1308; Merck Millipore, 1:100), Lys48-specific ubiquitin (Apu2; 05-1307; Merk Millipore, 1:100), GFP (GF200; 04363-24; Nacalai Tesque, 1:100), and TOM20 (F-10; sc-17764; Santa Cruz Biotechnology, 1:100). anti-mouse or anti-rabbit IgG conjugated to AlexaFluor 488 and 594 (Invitrogen, #A32723, #A32742, #A32731, #A32740) were used as a secondary antibody for immunostaining. Cells were washed with PBS, fixed for 15 min with 4% PFA in PBS, permeabilized with 0.1 % Triton in PBS for 10 min, washed with PBS, and blocked at room temperature for 1 h with skim milk (5% skim milk, 2.5% goat serum, 2.5% donkey serum, and 0.1% gelatin in PBS) or BSA (2% BSA and 0.02% sodium azide in PBS). Cells were then probed at room temperature for 1 h with the primary antibody diluted in blocking solution, washed with PBS, and labeled with the secondary antibody. To visualize bacterial and cellular DNA, cells were stained with 4′,6-diamidino-2-phenylindole (DAPI; Dojindo). Confocal fluorescence micrographs were acquired with an FV1000 laser-scanning microscope (Olympus).

**Calcium measurement using Fura 2-AM.** HeLa cells ($3 \times 10^4$ cells/well) grown on 96-well plates were infected with GAS. To analyze intracellular Ca$^{2+}$ levels, cells were loaded with 5 μM Fura 2-AM (Dojindo) in loading buffer (20 mM HEPES, 115 mM NaCl, 5.4 mM KCl, 0.8 mM MgCl$_2$, 1.8 mM CaCl$_2$, 13.8 mM glucose, 1.25 mM probencid, 0.04% Pluronic F-127, pH 7.4) for 1 h, and cells were washed with PBS and replaced with recording buffer (20 mM HEPES, 115 mM NaCl, 5.4 mM KCl, 3.8 mM MgCl$_2$, 1.8 mM CaCl$_2$, 13.8 mM glucose, 1.25 mM probencid, pH 7.4). Signal intensities (340 and 380 nm excitation, 510 nm emission) were measured using a Wallac ARVO SX Multilabel Counter (Perkin Elmer).

**Statistical analysis.** Cells containing LC3 or other autophagy factors were quantified through direct visualization using a confocal microscope (Olympus). Unless otherwise indicated, 200 GAS-infected cells were examined in each experiment, and at least three independent experiments were performed for each trial. Values, including those plotted, represent the mean ± standard error of the mean (SEM). Pearson's coefficient or Mander's M2 coefficients were calculated using the ImageJ software JACoP plugin with manually set thresholds[62]. Western blotting and immunoprecipitation experiments were repeated at least three times and representative blots are shown. Data were tested by two-tailed Student's *t* test, and a $P < 0.05$ was considered statistically significant and marked as *$P < 0.05$, **$P < 0.01$, ***$P < 0.001$, and NS for not significant.

**Reporting summary.** Further information on research design is available in the Nature Research Reporting Summary linked to this article.

## Data availability
The authors declare that the data supporting the findings of this study are available within the article and its Supplementary Information Files. All other relevant data supporting the findings of this study are available on request. Raw data for the figures are provided in the Source Data file.

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

## Acknowledgements

The authors acknowledge financial support from Grants-in-Aid for Scientific Research (16H05188, 15K15130, 26462776, 17K19552), the Takeda Science Foundation, the Research Program on Emerging and Re-emerging Infectious Diseases (18fk0108044h0202, 18fk0108073h0001) and J-PRIDE (18fm0208030h0002) from Japan Agency for Medical Research and Development, AMED.

## Author contributions

T.N. and I.N. conceived the study and provided reagents. T.N., S.N., H.T., K.M. and I.N. wrote the paper. T.N., S.S., A.M.-N., H.T. and C.A. performed the experiments and analyzed the data.

## Competing interests

The authors declare no competing interests.
