## [Peer Review File · Nature Communications]

Reviewers' comments:

Reviewer #1 (Remarks to the Author):

Manuscript entitled "TBC1D9 promotes 1 TBK1 activation through Ca²⁺ signaling in selective autophagy". This study appears to be of interest and is relatively convincing, hence the recommendation is a major revision.

In this manuscript Nozawa et al., intended to explore the role of a Ca²⁺ binding protein TBC1D9 in TBK1 activation and consequent selective autophagy (xenophagy and mitophagy). Interestingly, TBK1 activation shown here is independent of STING, a previously reported activator of TBK1 during infection. Authors screened several TBC/RabGAPs during Group A strep infection and found that TBC1D9 affects autophagy activation. Furthermore, TBC1D9 interacts with TBK1 and KO of TBC1D9 reduced phosphorylation of TBK1 and recruitment of pTBK1 and TBK1 effectors p62, NDP52 and LC3 to bacteria. Authors also show that TBC1D9 regulates mitophagy and that Ca²⁺ signaling is required for recruitment of TBC1D9 to mitochondria and consequent mitophagy. Overall, this is very interesting study identifying the role of TBC1D9 in microbial autophagy and mitophagy, the finding that TBK1 can be activated by means other than its published interactors STING, MAVS, TRIF, STX17, etc, the data presented in this study are clear and experiments in most parts are well controlled. There are however several issues that need to be addressed before this manuscript is acceptable for publication.

1. What do authors suppose is happening after TBK1 activation? Please see points 9-12 below, for the missed mechanisms and references for recently identified autophagy processes that depend on TBK1 that are completely (hopefully unintentionally) ignored here.
2. Authors seem to contradict their statement in data shown in Fig 3d where they show that TBC1D9 recruitment is independent of ATG5. Does this mean that this process is independent of autophagy?
3. Alternatively, the xenophagy reported here is dependent on autophagy which is independent of ATG5?
4. Is LC3 recruitment also independent of ATG5?
5. Given that this study is mainly focused on autophagy (xenophagy and mitophagy), authors should firm their results using KO/KD of earliest effectors of autophagy such as ULK complex, which has recently been shown to regulate xenophagy and mitophagy (Mol Cell. 2019 Apr 18;74(2):320-329; Mol Cell. 2019 Apr 18;74(2):347-362.), to test if this complex is required for recruitment of TBC1D9 to bacteria.
6. Figure 4m: authors need to replace the images or show all channels as the difference between control and IP3Rs samples is not apparent, the representative images should match the data shown in graphs.
7. Authors should also test if BAPTA-AM reduces colocalization of LC3 and TBC1D9 on bacteria.
8. Authors need to show quantifications related to figure 5d and 5e and should do some additional assay such as mitochondrial DNA quantifications or degradation of COXII (Nature 524, 309-314 (2015) to measure mitophagy.
9. Authors seem to be ignoring the published literature related to their study: Authors show that TBK1 phosphorylation/activation by GAS infection was independent of STING, which looks convincing but authors need to discuss the published literature related to STING and TBK1 in response to dsDNA (Proc Natl Acad Sci U S A. 2009 Dec 8; 106(49): 20842–20846).
10. The authors should discuss latest preprint published (<https://www.biorxiv.org/content/10.1101/634519v1>) related to role of TBK1 in phosphorylation of LC3s which would nicely fit with the data showing effect of TBC1D9 KO on recruitment of LC3 in Fig 1f. There is nothing gained by ignoring published literature.
11. Authors should also discuss recent papers showing role of TBK1 in autophagosome formation (Dev Cell. 2019 Apr 8;49(1):130-144; Mol Cell. 2019 Apr 18;74(2):347-362.).
12. Authors also did not discuss and appear to ignore (again, the message here is that there is nothing gained by ignoring established work) some important papers showing role of Galectin in

endomembrane damage (Nature. 2012; 482: 414-418) and a paper showing role of galectin3 in lysophagy (Dev Cell. 2016 Oct 10;39(1):13-27.)

Reviewer #2 (Remarks to the Author):

In this manuscript, Nozawa et al report about the importance of Ca²⁺ signalling in activation of TBK1 kinase during xenophagy and mitophagy. TBK1 has previously been shown to be recruited to Salmonella, leading to activation of xenophagy, thus restricting its proliferation. The authors have shown that TBK1 interacts with TBC1D9, which is recruited to Group A Streptococcus (GAS) or damaged mitochondria via Ca²⁺-dependent ubiquitin binding. The research performed by Nozawa et al sheds light on the mechanism underlying the recruitment of autophagosomal machinery to GAS, with strong biochemical data to support it. For the publication, the authors should address the functional relevance of this mechanism, outlined in the comments below, which is the main part lacking in this manuscript.

1. The authors throughout the manuscript predominantly use the stable Hela EmGFP-TBC1D9 cell line. In Figures 3c and 5a, the authors used an antibody which appears to stain the endogenous protein very nicely. What is missing is a control, as the authors do not show endogenous TBC1D9 under normal conditions, non-infected (Figure 3c) and untreated (Figure 5a). The authors should also show the endogenous TBC1D9 staining with and without Parkin, as well as with and without BAPTA-AM in Figure 5. Another point is that all images in Figure 5 are upon AO treatment. It would be good to include the untreated images, at least in supplementary material, and see if there is any recruitment of TBC1D9 to mitochondria as part of basal mitophagy.
2. The authors concluded that TBC1D9 and Ca²⁺ signalling are important for the activation of xenophagy and mitophagy. While it is clear that the autophagy machinery recruitment to GAS and damaged mitochondria is inhibited upon depletion of TBC1D9, or Ca²⁺ (by using BAPTA-AM), the authors have not performed any functional experiments to show the biological significance of the mechanism they revealed in this manuscript. The authors should include some functional assays to measure mitophagy and xenophagy, possibly even bacterial proliferation.
3. Figure 3e, delta2-722 is missing EF-hand domain.
4. Figure 3f, the last image is labelled with delta919-1266, whereas the text says it is delta926-1266. This is confusing, and it should be corrected.
5. Figure 3j, the authors should, if possible, include the full-length GST-TBC1D9, or explain in case they had issues purifying it.
6. Not essential, but it would be good to label Dapi in grayscale, to increase the contrast between the DNA labelling and the background.
7. Supplementary Figure 4c, add Pearson's coefficient of 'p62/Galectin-3 colocalization', as done in Supplementary Figure 4e.
8. Supplementary Figure 4g, the graph is not labelled.
9. Line 272, 'Figure 6a' missing.
10. Discussion is full of spelling mistakes and typos. The whole manuscript should be proofread before publication.
11. Some blots have kDa labels and some don't. It would be best to include the molecular weight next to the bands shown.

Reviewer #3 (Remarks to the Author):

The paper prepared by Nakagawa and team members reveals a novel calcium-dependent mechanism by which pathogens entering the cytosol of mammalian cells are removed by

xenophagy. The authors show that TBC1D9 is recruited to invading Group A Streptococcus infection in a calcium and ubiquitin-dependent manner. This provokes "activation" of TBK1 through phosphorylation and phagophore formation around the bacteria. This is a completely novel mechanism. Furthermore, the authors have expanded the relevance of their findings to mitophagy, which provides a more general cell biological implications. Until now, not much was known about the protagonists (TBC1D9 and TBK1) revealed in this work and the link towards calcium signaling is completely novel.

Overall, the work is novel and exciting. I enjoyed reading this paper. The experiments and results are well described and underpinned by a plethora of approaches, mainly at the cell biological level.

Some aspects though, particularly related to calcium signaling, require further scrutiny and study to underpin (some of) the conclusions.

Comments

1. Wild-type GAS provokes a cytosolic calcium rise that appears to be dependent on the presence of slo in the bacteria. The calcium rise is anticipated to originate from intracellular Ca²⁺ stores, but evidence is scant. Moreover, wild-type GAS appears to provoke Ca²⁺ rise that are even somewhat higher than A23187, which is due to release of ER but also from calcium fluxes across the plasma membrane. Thus, further work is needed:
 - a) In addition to the snap shots taken at 1, 2 or 4 hours, it could be informative to monitor live-cell calcium dynamics of wild-type GAS at much higher time resolution. (suggestion)
 - b) It will instrumental to use ratiometric calcium indicators such Fura-2 in order to rule out changes in cell volume or other artefacts occurring upon wild-type GAS infection.
 - c) The authors should demonstrate the effect of wild-type GAS on cytosolic Ca²⁺ levels in the absence of extracellular Ca²⁺ (thus using extracellular BAPTA) as well as in conditions of chelating intracellular Ca²⁺ (thus using BAPTA-AM). This is often omitted in studies, but an important control. Furthermore, it will be critical to assess the effect of IP3R knockdown on the cytosolic calcium levels in response to wild-type GAS. The same is true for 2-APB (see also below) and other inhibitors that should be tested. The use of PLC inhibitor U73122 (and its inactive enantiomer) could shed light on the necessity of PLC signaling. Moreover, the data with EGTA or EGTA-AM are puzzling on TBK-phosphorylation and also require further analysis.
 - d) There is no statistical analysis presented of the Ca²⁺ data presented in Figure 4A.
2.
 - a) The interpretation of the BAPTA-AM data to indicate the involvement of Ca²⁺ has to be taken with care, as BAPTA-AM treatments can also affect the Na/K ATPases. Besides validating that BAPTA-AM can reduce wild-type GAS-induced calcium rises, the authors should assess the effect of low-affinity Ca²⁺ chelators BAPTA-AM variants such dibromo- or difluoro-BAPTA-AM on wild-type GAS-induced TBK1 phosphorylation. Also, the use of EGTA-AM is needed (see also point 2b); EGTA-AM is a slow calcium chelator that is cell-permeable, but given the slow kinetics of the Ca²⁺ rise, this should work.
 - b) The authors should clarify whether they have used EGTA (as described in the results) or EGTA-AM (as indicated in the panel).
 - c) Unfortunately, 2-APB is not a selective IP3R inhibitor. In fact, 2-APB also inhibits SERCA and depletes ER calcium stores. This is not per se a problem, but will affect interpretation of the data. As such, wild-type GAS could fail to provoke a calcium rise, simply because ER Ca²⁺ stores were depleted. This should be further scrutinized using xestospongin B or C (but is less selective than XeB) and the use of thapsigargin, an irreversible SERCA inhibitor. The use of U73122 as PLC inhibitor could provide insight in the role of this enzyme.
 - d) It would be advisable to underpin these data with the overexpression of Ca²⁺-buffering proteins such as parvalbumin or calbindin D-28k.
3. It should be validated that wild-type GAS-induced calcium rise is independent of downstream components such as STING or TBC1D9 or TBK1 by using the available knockout cell models.
4. It would be interesting to assess whether the xenophagy of delta slo GAS could be restored

upon eliciting calcium signaling through artificial ways (such as A23187 exposure)?

5. It should be validated whether or not the Rab35 recruitment to GAS is dependent on cytosolic Ca²⁺ increases.

6. What is the consequence of failed xenophagy in case of GAS infection? What is the cell fate in experimental conditions with unresolved xenophagy upon GAS infection (such as TBC1D9 KO's; IP3R KO/KD cells).

7. The role of TBK1 phosphorylation in self-association requires further study. In other words, could TBK1-S172E (a phospho-mimic variant) self-assemble with TBK1 (wild-type or S172E mutant) in the absence of TBC1D9?

8. TBK1 phosphorylation is particularly affected by the absence of TBC1D9 at 4 hours, while the effect at 2 hours is much less evident. Moreover, there remains some TBK1 phosphorylation in the absence of TBC1D9. How can this be explained? Is still inhibited upon cytosolic calcium buffering / IP3R knockdown. In other words, is the effect of TBC1D9 KO and calcium chelation/IP3R KD on TBK1 phosphorylation additive?

9. It should be clarified which IP3R isoform has been knocked down. Indeed, IP3Rs are not a protein as such, but are expressed as three different isoforms (IP3R1, IP3R2 and IP3R3).

Geert Bultynck

Reviewer #1 (Remarks to the Author):

Comment #1-1)

Manuscript entitled "TBC1D9 promotes 1 TBK1 activation through Ca²⁺ signaling in selective autophagy". This study appears to be of interest and is relatively convincing, hence the recommendation is a major revision. In this manuscript Nozawa et al., intended to explore the role of a Ca²⁺ binding protein TBC1D9 in TBK1 activation and consequent selective autophagy (xenophagy and mitophagy). Interestingly, TBK1 activation shown here is independent of STING, a previously reported activator of TBK1 during infection. Authors screened several TBC/RabGAPs during Group A strep infection and found that TBC1D9 affects autophagy activation. Furthermore, TBC1D9 interacts with TBK1 and KO of TBC1D9 reduced phosphorylation of TBK1 and recruitment of pTBK1 and TBK1 effectors p62, NDP52 and LC3 to bacteria. Authors also show that TBC1D9 regulates mitophagy and that Ca²⁺ signaling is required for recruitment of TBC1D9 to mitochondria and consequent mitophagy. Overall, this is very interesting study identifying the role of TBC1D9 in microbial autophagy and mitophagy, the finding that TBK1 can be activated by means other than its published interactors STING, MAVS, TRIF, STX17, etc, the data presented in this study are clear and experiments in most parts are well controlled. There are however several issues that need to be addressed before this manuscript is acceptable for publication.

Response to comment #1-1)

First, we wish to thank the reviewer for these positive comments on our study. We have performed additional experiments to clarify some concepts, as described below.

Comment #1-2)

What do authors suppose is happening after TBK1 activation? Please see points 9-12 below, for the missed mechanisms and references for recently identified autophagy processes that depend on TBK1 that are completely (hopefully unintentionally) ignored here.

Response to comment #1-2)

As suggested by the reviewer, recent advances have revealed that TBK1 activation leads to various downstream signaling molecules such as p62, NDP52, STX17, RAB7, and the ULK1 complex. In the original manuscript, as we focused on the activation mechanism of TBK1 during xenophagy, we did not mention the processes depending on TBK1.

However, as suggested by reviewers, we agree that the importance of the autophagy process regulated by TBK1 that is activated through TBC1D9 and Ca²⁺ signaling needs to be emphasized. Thus, we have cited the relevant references and performed additional experiments to reveal which autophagy process is influenced by TBK1 activation downstream of TBC1D9 recruitment. We previously showed that TBK1 activation leads to NDP52 and LC3 recruitment to GAS. Moreover, recent reports have revealed that TBK1 and NDP52 regulate ULK1 complex recruitment to bacteria in xenophagy. We then examined the localization of ULK1 complex during GAS infection. As shown in Fig. 2e, ULK1 was recruited to ubiquitin-coated GAS and this recruitment was suppressed by TBC1D9 knockout. We also observed that ULK1 and its complex components (ULK2, FIP200, and ATG13) were required for xenophagic degradation of GAS. Therefore, the results suggest that TBC1D9-mediated TBK1 activation leads to ULK1 complex recruitment and initiation of xenophagy.

Comment #1-3)

Authors seem to contradict their statement in data shown in Fig 3d where they show that TBC1D9 recruitment is independent of ATG5. Does this mean that this process is independent of autophagy?

Response to comment #1-3)

We believe that the recruitment of TBC1D9 to bacteria is an ATG5-independent mechanism whereas the TBC1D9-mediated xenophagy process is ATG5-dependent. In Fig. 3d of the original manuscript, we observed TBC1D9-positive GAS in both wild-type and ATG5-knockout cells. This suggested that TBC1D9 can be recruited to bacteria prior to LC3 recruitment. However, TBC1D9 recruitment efficiency in ATG5-knockout cells was significantly lower than that in wild-type cells, indicating that LC3 might promote the recruitment of TBC1D9. Taken together, TBC1D9 recruitment to GAS via ubiquitin and Ca²⁺-binding might occur prior to ATG5-mediated LC3 recruitment (autophagic membrane formation), and LC3 facilitates TBC1D9 accumulation around GAS via an unknown mechanism. During mitophagy, autophagy adaptors such as NDP52 and Optineurin are first recruited to ubiquitin-coated mitochondria, and following LC3 recruitment they provide positive feedback to recruit autophagy adaptors (Padman et al., Nat Commun. 2019); thus, the TBC1D9-TBK1-autophagy adaptors axis and LC3 might amplify autophagy signaling at the targeting sites during selective autophagy.

Comment #1-4)

Alternatively, the xenophagy reported here is dependent on autophagy which is independent of ATG5?

Response to comment #1-4)

Xenophagy during GAS infection is ATG5-dependent autophagy (Oda et al., PLoS One. 2016; Minowa-Nozawa et al., EMBO J. 2017).

Comment #1-5)

Is LC3 recruitment also independent of ATG5?

Response to comment #1-5)

Since LC3 recruitment to GAS is not observed in ATG5-knockout HeLa cells, LC3 recruitment is ATG5-dependent (Oda et al., PLoS One. 2016).

Comment #1-6)

Given that this study is mainly focused on autophagy (xenophagy and mitophagy), authors should firm their results using KO/KD of earliest effectors of autophagy such as ULK complex, which has recently been shown to regulate xenophagy and mitophagy (Mol Cell. 2019 Apr 18;74(2):320-329; Mol Cell. 2019 Apr 18;74(2):347-362.), to test if this complex is required for recruitment of TBC1D9 to bacteria.

Response to comment #1-6)

As suggested by the reviewer, we examined the involvement of the ULK1 complex in TBC1D9 recruitment in the revised experiments. We knocked down the expression of FIP200 and ULK1 using siRNA, and observed the localization of TBC1D9 during GAS infection. Recruitment of TBC1D9 to ubiquitin-positive GAS was not affected by knockdown of FIP200 or ULK1 (below figure). Alternatively, we found that ULK1-positive GAS was significantly decreased in TBC1D9-knockout cells (Fig. 2 e, f). Recruitment of ATG13 to depolarized mitochondria was also suppressed by TBC1D9 knockout (Fig. 6 d,e). These observations suggest that TBC1D9 is an upstream molecule of the ULK1 complex during the xenophagy and mitophagy initiation process.

Comment #1-7)

Figure 4m: authors need to replace the images or show all channels as the difference between control and IP3Rs samples is not apparent, the representative images should match the data shown in graphs.

Response to comment #1-7)

As suggested, we showed all channels and included insets of LC3 images in the control and IP3R-knockdown cells.

Comment #1-8)

Authors should also test if BAPTA-AM reduces colocalization of LC3 and TBC1D9 on bacteria.

Response to comment #1-8)

We have examined the effects of BAPTA-AM on the colocalization of LC3 and TBC1D9 in bacteria. As expected, colocalization was significantly decreased by BAPTA-AM (Fig. 4d).

Comment #1-9)

Authors need to show quantifications related to figure 5d and 5e and should do some additional assay such as mitochondrial DNA quantifications or degradation of COXII (Nature 524, 309-314 (2015) to measure mitophagy.

Response to comment #1-9)

As suggested by the reviewer, to evaluate mitophagy activity, we quantified COXII degradation in TBC1D9 KO cells. As shown in Fig. 6f and 6g, AO treatment-induced COXII degradation was diminished in TBC1D9 KO cells, indicating that TBC1D9 is required for efficient mitophagy.

Comment #1-10)

Authors seem to be ignoring the published literature related to their study: Authors show that TBK1 phosphorylation/activation by GAS infection was independent of STING, which looks convincing but authors need to discuss the published literature related to STING and TBK1 in response to dsDNA (Proc Natl Acad Sci U S A. 2009 Dec 8; 106(49): 20842–20846).

Response to comment #1-10)

As suggested, we agree that these references are important to discuss TBK1 activation in autophagy. We have thus cited these references in the revised manuscript.

Comment #1-11)

The authors should discuss latest preprint published (<https://www.biorxiv.org/content/10.1101/634519v1>) related to role of TBK1 in phosphorylation of LC3s which would nicely fit with the data showing effect of TBC1D9 KO on recruitment of LC3 in Fig 1f. There is nothing gained by ignoring published literature.

Response to comment #1-11)

As suggested, we have cited this reference and discussed it in the revised manuscript.

Comment #1-12)

Authors should also discuss recent papers showing role of TBK1 in autophagosome formation (Dev Cell. 2019 Apr 8;49(1):130-144; Mol Cell. 2019 Apr 18;74(2):347-362.).

Response to comment #1-12)

As suggested, we have cited this and discussed it in the revised manuscript.

Comment #1-13)

Authors also did not discuss and appear to ignore (again, the message here is that there is nothing gained by ignoring established work) some important papers showing role of Galectin in endomembrane damage (Nature. 2012; 482: 414-418) and a paper showing role of galectin3 in lysophagy (Dev Cell. 2016 Oct 10;39(1):13-27.)

Response to comment #1-13)

As suggested, we have added to the discussion in the revised manuscript. Specifically, since we showed that ULK1 complex recruitment is regulated by the TBC1D9-TBK1 axis in xenophagy, we discussed the difference in xenophagy and lysophagy.

Reviewer #2 (Remarks to the Author):

Comment #2-1)

In this manuscript, Nozawa et al report about the importance of Ca²⁺ signalling in activation of TBK1 kinase during xenophagy and mitophagy. TBK1 has previously been shown to be recruited to Salmonella, leading to activation of xenophagy, thus restricting its proliferation. The authors have shown that TBK1 interacts with TBC1D9, which is recruited to Group A Streptococcus (GAS) or damaged mitochondria via Ca²⁺-dependent ubiquitin binding. The research performed by Nozawa et al sheds light on the mechanism underlying the recruitment of autophagosomal machinery to GAS, with strong biochemical data to support it. For the publication, the authors should address the functional relevance of this mechanism, outlined in the comments below, which is the main part lacking in this manuscript.

Response to comment #2-1)

We first wish to thank the reviewer for the positive comments on our study. We have performed additional experiments to clarify some concepts, as described below.

Comment #2-2)

The authors throughout the manuscript predominantly use the stable Hela EmGFP-TBC1D9 cell line. In Figures 3c and 5a, the authors used an antibody which appears to stain the endogenous protein very nicely. What is missing is a control, as the authors do not show endogenous TBC1D9 under normal conditions, non-infected (Figure 3c) and untreated (Figure 5a). The authors should also show the endogenous TBC1D9 staining with and without Parkin, as well as with and without BAPTA-AM in Figure 5. Another point is that all images in Figure 5 are upon AO treatment. It would be good to include the untreated images, at least in supplementary material, and see if there is any recruitment of TBC1D9 to mitochondria as part of basal mitophagy.

Response to comment #2-1)

As suggested, we showed endogenous TBC1D9 localization in control cells (Fig. 3c and 5a). We also exhibited the subcellular localization of endogenous TBC1D9 with and without Parkin (Supplementary Fig. 9a). In addition, we showed TBC1D9 mutants localization under normal conditions (Supplementary Fig. 9b). As the reviewer commented, TBC1D9 partially colocalized with mitochondria even in the control condition. Thus, TBC1D9 might also be involved in basal mitophagy.

Comment #2-3)

The authors concluded that TBC1D9 and Ca²⁺ signalling are important for the activation of xenophagy and mitophagy. While it is clear that the autophagy machinery recruitment to GAS and damaged mitochondria is inhibited upon depletion of TBC1D9, or Ca²⁺ (by using BAPTA-AM), the authors have not performed any functional experiments to show the biological significance of the mechanism they revealed in this manuscript. The authors should include some functional assays to measure mitophagy and xenophagy, possibly even bacterial proliferation.

Response to comment #2-3)

As suggested, to investigate the functional significance of TBC1D9 in xenophagy and mitophagy, we assessed the degradation of GAS and mitochondria using TBC1D9-knockout cells. In Fig. 2g of the revised manuscript, survival GAS in TBC1D9 knockout HeLa cells at 6 h after infection was significantly higher than that in wild-type HeLa cells, suggesting that degradation of GAS is diminished by TBC1D9 knockout in HeLa cells. Moreover, AO-treatment-induced COXII degradation was also suppressed in TBC1D9 KO cells (Fig. 6f and 6g). Therefore, we concluded that TBC1D9 is functionally important for xenophagy and mitophagy.

Comment #2-4)

Figure 3e, delta2-722 is missing EF-hand domain.

Response to comment #2-4)

We appreciate your suggestion. We have modified Figure 3e in the revised manuscript.

Comment #2-5)

Figure 3f, the last image is labelled with delta919-1266, whereas the text says it is delta926-1266. This is confusing, and it should be corrected.

Response to comment #2-5)

We apologize for this error; we have modified this label to “delta926-1266”.

Comment #2-6)

Figure 3j, the authors should, if possible, include the full-length GST-TBC1D9, or explain in case they had issues purifying it.

Response to comment #2-6)

Our attempt to construct full-length GST-TBC1D9 failed. We then used GST-TBC1D9 aa926-1100, the region responsible for ubiquitin binding. To examine the interaction between full-length TBC1D9 and K63-ubiquitin, we performed a PLA assay and showed that full-length TBC1D9 could also associate with K63-ubiquitin (Fig. 3k).

Comment #2-7)

Not essential, but it would be good to label Dapi in grayscale, to increase the contrast between the DNA labelling and the background.

Response to comment #2-7)

As suggested, to increase the contrast of DNA labeling, we have shown DAPI in magenta or cyan in the revised manuscript.

Comment #2-8)

Supplementary Figure 4c, add Pearson's coefficient of 'p62/Galectin-3 colocalization', as done in Supplementary Figure 4e.

Response to comment #2-8)

As suggested, we have added the label 'p62/Galectin-3 colocalization' to the revised manuscript.

Comment #2-9)

Supplementary Figure 4g, the graph is not labelled.

Response to comment #2-9)

We have labelled the graph in the revised supplementary Figure 4g.

Comment #2-10)

Line 272, 'Figure 6a' missing.

Response to comment #2-10)

We have added the Figure label in the revised manuscript.

Comment #2-11)

Discussion is full of spelling mistakes and typos. The whole manuscript should be proofread before publication.

Response to comment #2-11)

As suggested, we have proofread the entire manuscript in the revised submission.

Comment #2-12)

Some blots have kDa labels and some don't. It would be best to include the molecular weight next to the bands shown.

Response to comment #2-12)

As suggested, we have shown kDa labels in western blot images.

Reviewer #3 (Remarks to the Author):

Comment #3-1)

The paper prepared by Nakagawa and team members reveals a novel calcium-dependent mechanism by which pathogens entering the cytosol of mammalian cells are removed by xenophagy. The authors show that TBC1D9 is recruited to invading Group A Streptococcus infection in a calcium and ubiquitin-dependent manner. This provokes “activation” of TBK1 through phosphorylation and phagophore formation around the bacteria. This is a completely novel mechanism. Furthermore, the authors have expanded the relevance of their findings to mitophagy, which provides a more general cell biological implications. Until now, not much was known about the protagonists (TBC1D9 and TBK1) revealed in this work and the link towards calcium signaling is completely novel.

Overall, the work is novel and exciting. I enjoyed reading this paper. The experiments and results are well described and underpinned by a plethora of approaches, mainly at the cell biological level.

Some aspects though, particularly related to calcium signaling, require further scrutiny and study to underpin (some of) the conclusions.

Response to comment #3-1)

We appreciate your positive comments on our study and your advice. In response to all the comments, we have performed the suggested experiments and included data in the revised submission.

Comment #3-2a)

Wild-type GAS provokes a cytosolic calcium rise that appears to be dependent on the presence of slo in the bacteria. The calcium rise is anticipated to originate from intracellular Ca²⁺ stores, but evidence is scant. Moreover, wild-type GAS appears to provoke Ca²⁺ rise that are even somewhat higher than A23187, which is due to release of ER but also from calcium fluxes across the plasma membrane. Thus, further work is needed:

In addition to the snap shots taken at 1, 2 or 4 hours, it could be informative to monitor live-cell calcium dynamics of wild-type GAS at much higher time resolution.

(suggestion)

Response to comment #3-2a)

We agree that live-cell imaging could be informative to monitor the calcium dynamics during infection; however, unfortunately we could not perform this experiment in our research environment.

Comment #3-2b)

It will instrumental to use ratiometric calcium indicators such Fura-2 in order to rule out changes in cell volume or other artefacts occurring upon wild-type GAS infection.

Response to comment #3-2b)

As suggested, we used Fura 2-AM to monitor the Ca^{2+} dynamics during GAS infection. Time course of the Fura 2-AM ratio during GAS infection revealed that cytosolic Ca^{2+} increased from 2 h after infection and that this elevation was SLO-dependent (Fig. 4a). These results correspond with the data obtained using G-GaMP3 (Supplementary Fig. 5a and 5b).

Comment #3-2c)

The authors should demonstrate the effect of wild-type GAS on cytosolic Ca^{2+} levels in the absence of extracellular Ca^{2+} (thus using extracellular BAPTA) as well as in conditions of chelating intracellular Ca^{2+} (thus using BAPTA-AM). This is often omitted in studies, but an important control. Furthermore, it will be critical to assess the effect of IP3R knockdown on the cytosolic calcium levels in response to wild-type GAS. The same is true for 2-APB (see also below) and other inhibitors that should be tested. The use of PLC inhibitor U73122 (and its inactive enantiomer) could shed light on the necessity of PLC signaling. Moreover, the data with EGTA or EGTA-AM are puzzling on TBK-phosphorylation and also require further analysis.

Response to comment #3-2b)

As suggested, we first examined the Ca^{2+} elevation during GAS infection in BAPTA-AM and BAPTA treated conditions using Fura 2-AM. We found that BAPTA partially suppressed Ca^{2+} elevation, and that BAPTA-AM effectively inhibited Ca^{2+} mobilization (Supplementary Fig. 5f and 5g). We also quantified TBK1 activation in cells with BAPTA-AM, BAPTA, EGTA-AM, and EGTA. As shown in Supplementary Fig. 5f and 5g, EGTA-AM and EGTA did not influence TBK1 activation, whereas BAPTA-AM clearly suppressed TBK1 activation in a dose-dependent manner. BAPTA affected TBK1 activation only slightly. Collectively, although extracellular Ca^{2+} might also be involved

in TBK1 activation, intracellular Ca^{2+} would be critical for xenophagy during GAS infection.

In addition, we used the PLC inhibitor U731122 to test whether PLC signaling is involved in Ca^{2+} -mediated TBK1 activation and found that U73122 significantly inhibited Ca^{2+} elevation and TBK1 activation during GAS infection (Supplementary Fig. 5f and 5g). These results support that IP3 signaling is involved in Ca^{2+} mobilization and TBK1 activation during GAS infection for xenophagy.

Comment #3-2c)

There is no statistical analysis presented of the Ca^{2+} data presented in Figure 4A.

Response to comment #3-2c)

We performed the statistical analysis and included the data in Supplementary Fig. 5b.

Comment #3-3a)

The interpretation of the BAPTA-AM data to indicate the involvement of Ca^{2+} has to be taken with care, as BAPTA-AM treatments can also affect the Na/K ATPases. Besides validating that BAPTA-AM can reduce wild-type GAS-induced calcium rises, the authors should assess the effect of low-affinity Ca^{2+} chelators BAPTA-AM variants such dibromo- or difluoro-BAPTA-AM on wild-type GAS-induced TBK1 phosphorylation. Also, the use of EGTA-AM is needed (see also point 2b); EGTA-AM is a slow calcium chelator that is cell-permeable, but given the slow kinetics of the Ca^{2+} rise, this should work.

Response to comment #3-3a)

We appreciate your informative advice about calcium inhibitors. We used dibromo-BAPTA-AM and quantified TBK1 activation in response to GAS infection. Treatment with dibromo-BAPTA-AM suppressed TBK1 activation, but was only mildly-effective compared to BAPTA-AM (Supplementary Fig. 5f and 5g). Also, we have added the data for EGTA-AM in the revised manuscript. We did not observe any effect of EGTA-AM on TBK1 activation during GAS infection. We have mentioned this result in the revised manuscript.

Comment #3-3b)

The authors should clarify whether they have used EGTA (as described in the results) or EGTA-AM (as indicated in the panel).

Response to comment #3-3b)

We made an error in the label in Fig. 4h of the original manuscript. In the original manuscript, we showed the data using EGTA (not EGTA-AM). As described above, we have added the results of both EGTA-AM and EGTA.

Comment #3-3c)

Unfortunately, 2-APB is not a selective IP3R inhibitor. In fact, 2-APB also inhibits SERCA and depletes ER calcium stores. This is not per se a problem, but will affect interpretation of the data. As such, wild-type GAS could fail to provoke a calcium rise, simply because ER Ca²⁺ stores were depleted. This should be further scrutinized using xestospongin B or C (but is less selective than XeB) and the use of thapsigargin, an irreversible SERCA inhibitor. The use of U73122 as PLC inhibitor could provide insight in the role of this enzyme.

Response to comment #3-3c)

We investigated the effects of these inhibitors on TBK1 activation during wild-type GAS infection, and found that in our experiment condition, XeC and U73122 significantly inhibited TBK1 activation whereas thapsigargin did not. These results suggest that IP3R-mediated calcium release is involved in TBK1 activation during GAS infection.

Comment #3-3d)

It would be advisable to underpin these data with the overexpression of Ca²⁺-buffering proteins such as parvalbumin or calbindin D-28k.

Response to comment #3-3d)

As suggested, we overexpressed parvalbumin (PVALB) and calbindin D-28k (CALB), and examined the activation of TBK1 during GAS infection. TBK1 phosphorylation was clearly diminished in PVALB or CALB expressing cells (Supplementary Fig. 6b). Moreover, recruitment of TBC1D9 to ubiquitin-positive GAS was also suppressed by overexpression of PVALB or CALB (Supplementary Fig. 6a). These results support that cytosolic Ca²⁺ is critical for TBC1D9 recruitment and TBK1 activation against GAS infection.

Comment #3-4)

It should be validated that wild-type GAS-induced calcium rise is independent of downstream components such as STING or TBC1D9 or TBK1 by using the available knockout cell models.

Response to comment #3-4)

As suggested by the reviewer, we examined the Ca^{2+} level in STING, TBC1D9, and TBK1 knockout cells during wild-type GAS infection, and observed that Ca^{2+} elevation was observed even in these knockout cells (Supplementary Fig. 5i).

Comment #3-5)

It would be interesting to assess whether the xenophagy of delta slo GAS could be restored upon eliciting calcium signaling through artificial ways (such as A23187 exposure)?

Response to comment #3-5)

We infected HeLa cells with GAS Δslo and treated them with A23187 to elicit calcium signaling, and then examined xenophagy. None of TBC1D9, p-TBK1, and LC3 was recruited to GAS Δslo even when with A23187 treatment (below figures). Although Ca^{2+} elevation in the cytosol is critical for TBC1D9-mediated TBK1 activation, ubiquitin accumulation on bacteria is also essential for the TBC1D9-TBK1 axis and this ubiquitin recruitment is independent on Ca^{2+} signaling (Supplementary Fig. 5c and 5d). Therefore, we think that Ca^{2+} signaling and the ubiquitination process are activated in parallel in response to wild-type GAS infection, and that both events are required for TBC1D9-mediated xenophagy by promoting TBK1 activation.

Comment #3-6)

It should be validated whether or not the Rab35 recruitment to GAS is dependent on cytosolic Ca²⁺ increases.

Response to comment #3-6)

As suggested, we observed Rab35 localization during infection with BAPTA-AM. BAPTA-AM did not inhibit the recruitment of Rab35 to bacteria (supplementary Fig. 5e). This result is consistent with our previous results that Rab35 is recruited to GAS before cytosolic invasion of GAS.

Comment #3-7)

What is the consequence of failed xenophagy in case of GAS infection? What is the cell fate in experimental conditions with unresolved xenophagy upon GAS infection (such as TBC1D9 KO's; IP3R KO/KD cells).

Response to comment #3-7)

In the revised manuscript, we examined bacterial survival in several knockout or knockdown cells. Bacterial survival at 6 h after infection was significantly increased in TBC1D9 knockout cells as well as in ATG5 knockout cells (Fig. 2g; Minowa Nozawa et al. EMBO J), indicating that TBC1D9 is required for xenophagic degradation of GAS. We also found that knockdown of IP3Rs increased bacterial survival (Fig. 4k). Collectively, IP3Rs- and TBC1D9-mediated xenophagy was suggested to restrict bacterial proliferation.

Comment #3-8)

The role of TBK1 phosphorylation in self-association requires further study. In other words, could TBK1-S172E (a phospho-mimic variant) self-assemble with TBK1 (wild-type or S172E mutant) in the absence of TBC1D9?

Response to comment #3-8)

As suggested, we examined the self-association of TBK1-S172E in TBC1D9 knockout cells. Similar to wild-type TBK1, interaction between FLAG-TBK1-S172E and EmGFP-TBK1-S172E was still attenuated in TBC1D9 knockout cells. We therefore concluded that TBC1D9 is involved in the self-association of TBK1. We have added these results to Fig. 1i in the revised manuscript.

Comment #3-9)

TBK1 phosphorylation is particularly affected by the absence of TBC1D9 at 4 hours, while the effect at 2 hours is much less evident. Moreover, there remains some TBK1 phosphorylation in the absence of TBC1D9. How can this be explained? Is still inhibited upon cytosolic calcium buffering / IP3R knockdown. In other words, is the effect of TBC1D9 KO and calcium chelation/IP3R KD on TBK1 phosphorylation additive?

Response to comment #3-9)

In TBC1D9 knockout cells, the phosphorylated TBK1 (p-TBK1) level was significantly increased in normal conditions (the reason of this is not clear yet). Then, to evaluate the increase in p-TBK1 during infection, we quantified the p-TBK1 amount relative to 0 h.p.i in WT and knockout cells. As shown in Fig. 1b, increase of p-TBK1 relative to that before infection significantly decreased in TBC1D9 knockout cells at both 2 and 4 h. To validate this, we transiently knocked down the TBC1D9 expression using siRNA and found that the siRNA for TBC1D9 inhibited TBK1 phosphorylation even at 2 h after infection (Supplementary Fig. 7b, c). In addition, we found that the p-TBK1 level was comparable between TBC1D9-knockdown and TBC1D9/IP3R1 double-knockdown cells, suggesting that the effect of TBC1D9 knockdown and IP3Rs knockdown on TBK1 activation is not additive.

Comment #3-10)

It should be clarified which IP3R isoform has been knocked down. Indeed, IP3Rs are not a protein as such, but are expressed as three different isoforms (IP3R1, IP3R2 and IP3R3).

Response to comment #3-10)

In our original manuscript, we knocked down all isoforms (IP3R1, IP3R2, and IP3R3). To identify which IP3R isoforms is involved in xenophagy induction, we selectively knocked down the expression of each IP3R isoform. We found that IP3R1 knockdown most profoundly diminished TBK1 activation and bacterial degradation (Fig. 4f and 4k). However, IP3R2 and IP3R3 were also involved in TBK1-mediated xenophagy (Fig. 4f and 4k).

REVIEWERS' COMMENTS:

Reviewer #1 (Remarks to the Author):

The authors have addressed many comments by the reviewers.

What remains unattended by the authors, and is key to understanding these relationships are the following:

1) The role of a RabGAP is completely ignored here for its role as a Rab regulator. This seems to be necessary before one can accept the relationships and effects described by the authors. This is all the more important since the authors again fail to give credit for the role of TBC1D9 in autophagy (Longatti et al., 2012, JHCB, Fig. 4; <http://doi.org/10.1083/jcb.201111079>).

2) The authors have missed the opportunity given prior reviews to provide experimental information whether this RabGAP acts via a Rab and which one or alternatively that Rabs and RabGAP activities are not involved. This is not a trivial matter and has to be addressed.

3) The authors continue with their highly selective referencing and instead must reference seminal papers on xenophagy; not derivative subsequent studies.

4) The authors missed the opportunity to address experimentally and reference recent relationships shown regarding TBK and ULK (two Mol Cell papers) and TBK1 and Stx17 (a Dev Cell paper). They seem to choose to claim that "this is the first time to their knowledge"

Given the above, this reviewer cannot endorse this study in its present form, notwithstanding several specific improvements that are evident and acknowledged here.

Reviewer #2 (Remarks to the Author):

The authors have successfully addressed all raised points from the first review. I have no further comments.

Reviewer #3 (Remarks to the Author):

The authors have adequately revised the manuscript. The new results further strengthen the conclusions at different levels. This is a great study.

Two minor comments:

1. Inspecting the data, it seems that BAPTA-AM (a high affinity calcium chelator) is more potent in suppressing TBK1 phosphorylation than dibromo-BAPTA-AM (a low affinity calcium chelator). However, in the description of the results, the authors undermine the calcium effect as they mention that similarly to BAPTA-AM dibromo-BAPTA-AM is suppressing TBK1 phosphorylation, but this might mislead the reader who may worry that the effect of BAPTA-AM is not via calcium chelation. Hence, I would recommend to explicitly mention in the results that dibromo-BAPTA-AM was less effective in suppressing p-TBK1 than BAPTA-AM, consistent with a role for calcium signaling in the process.

2. The authors rightly mention about the adverse effects of BAPTA-AM on the Na/K ATPase, but do not provide a referenced source for this claim. I would recommend to cite the papers discovering

and discussing this issue: Smith NA et al, *Science Signaling*, 2018 and Bootman MD et al, *Cell Calcium*, 2018.

Reviewer #1 (Remarks to the Author):

The authors have addressed many comments by the reviewers.

What remains unattended by the authors, and is key to understanding these relationships are the following:

1) The role of a RabGAP is completely ignored here for its role as a Rab regulator. This seems to be necessary before one can accept the relationships and effects described by the authors. This is all the more important since the authors again fail to give credit for the role of TBC1D9 in autophagy (Longatti et al., 2012, JHCB, Fig. 4; <http://doi.org/10.1083/jcb.201111079>).

2) The authors have missed the opportunity given prior reviews to provide experimental information whether this RabGAP acts via a Rab and which one or alternatively that Rabs and RabGAP activities are not involved. This is not a trivial matter and has to be addressed.

We understand the importance of roles of RabGAP as a Rab regulator, because we also previously reported that TBC1D10A negatively regulate autophagy through Rab35 (Minowa-Nozawa et al., EMBO J. 2017). However, in Fig. 3f, we showed that TBC domain is not required for the recruitment of TBC1D9 to bacteria. In addition, we have data that GAP activity-deficient TBC1D9 (TBC1D9 R289A) fully rescued the recruitment of phosphorylated TBK1 to bacteria (below figure), demonstrating that TBC1D9 targets ubiquitin and regulates TBK1 activation in a GAP activity-independent manner. Therefore, we here focused on the role of TBC1D9 other than Rab regulator.

3) The authors continue with their highly selective referencing and instead must reference seminal papers on xenophagy; not derivative subsequent studies.

As suggested we have added some seminal paper on xenophagy.

4) The authors missed the opportunity to address experimentally and reference recent relationships

shown regarding TBK and ULK (two Mol Cell papers) and TBK1 and Stx17 (a Dev Cell paper).

They seem to chose to claim that "this is the first time to their knowledge"

In the previous revision, we experimentally verified the roles of TBK1 activation in ULK1 with citations. In two Mol Cell paper, it was shown that TBK1 and NDP52 recruit ULK1 complex to bacteria or mitochondria in xenophagy and mitophagy, respectively. We have cited these papers and examined whether TBC1D9-mediated TBK1 activation is also involved in the recruitment of ULK1 complex to bacteria or mitochondria. We also cited a Dev Cell paper and discussed in the Discussion section about the possibility that TBC1D9-mediated TBK1 activation regulate the assembly of ATG13 and FIP200.

Reviewer #3 (Remarks to the Author):

The authors have adequately revised the manuscript. The new results further strengthen the conclusions at different levels. This is a great study.

Two minor comments:

1. Inspecting the data, it seems that BAPTA-AM (a high affinity calcium chelator) is more potent in suppressing TBK1 phosphorylation than dibromo-BAPTA-AM (a low affinity calcium chelator). However, in the description of the results, the authors undermine the calcium effect as they mention that similarly to BAPTA-AM dibromo-BAPTA-AM is suppressing TBK1 phosphorylation, but this might mislead the reader who may worry that the effect of BAPTA-AM is not via calcium chelation. Hence, I would recommend to explicitly mention in the results that dibromo-BAPTA-AM was less effective in suppressing p-TBK1 than BAPTA-AM, consistent with a role for calcium signaling in the process.

We thank the reviewer for constructive advices. As suggested, we have modified our manuscript (page 14, lines 8-11).

2. The authors rightly mention about the adverse effects of BAPTA-AM on the Na/K ATPase, but do not provide a referenced source for this claim. I would recommend to cite the papers discovering and discussing this issue: Smith NA et al, Science Signaling, 2018 and Bootman MD et al, Cell Calcium, 2018.

We thank the reviewer for this advice. We have cited these papers in page 15 line 4.